# CurES: From Gradient Analysis to Efficient Curriculum Learning for Reasoning LLMs

**Yongcheng Zeng**[1,2,*]**, Zexu Sun**[3,*]**, Bokai Ji**[3]**, Erxue Min**[3]**, Hengyi Cai**[3]**,**
**Shuaiqiang Wang**[3]**, Dawei Yin**[3]**, Haifeng Zhang**[1,2,†]**, Xu Chen**[5,†]**, Jun Wang**[4,†]

[1]Institute of Automation, Chinese Academy of Sciences
[2]School of Artificial Intelligence, University of Chinese Academy of Sciences
[3]Baidu Inc. [4]University College London
[5]Gaoling School of Artificial Intelligence, Renmin University of China
`sunzexu0826@gmals.com`, `zengyongcheng2022@ia.ac.cn`

## Abstract

Curriculum learning plays a crucial role in enhancing the training efficiency of large language models (LLMs) on reasoning tasks. However, existing methods often fail to adequately account for variations in prompt difficulty or rely on simplistic filtering mechanisms to select prompt datasets within a narrow criterion range, resulting in significant computational waste. In this work, we approach the problem from the perspective of reinforcement learning gradient optimization, offering a systematic and theoretical investigation into how to improve the training efficiency of LLMs. We identify two key factors influencing training efficiency: the selection of training prompts and the allocation of rollout quantities across different prompts. Our theoretical analysis reveals that the sampling distribution of prompts dictates the convergence rate of gradient descent, while the allocation of the rollout quantity influences the consistency and stability of overall gradient updates. Based on these insights, we propose CurES, an efficient training method that accelerates convergence and employs Bayesian posterior estimation to minimize computational overhead. Experiments demonstrate that our CurES outperforms Group Relative Policy Optimization (GRPO) by **+3.30** points and **+4.82** points with 1.5B and 7B models, respectively, and exceeds the best prior sample efficient methods by **+2.12** points on average across eight math reasoning benchmarks. Additionally, CurES exhibits faster convergence compared to baselines, including GRPO.

 **GitHub:** https://github.com/ZexuSun/CurES

## 1 Introduction

Although Reinforcement Learning with Verifiable Reward (RLVR) (Guo et al., 2025; Lambert et al., 2024; Team et al., 2025) has emerged as a powerful paradigm for reasoning tasks of Large Language Models (LLMs), prevailing approaches often rely on uniform sampling strategies that treat all training instances identically (Zeng et al., 2025; Xie et al., 2025). This paradigm fails to account for the inherent heterogeneity in prompt difficulty and the varying training utility that different prompts offer. Consequently, computational resources are inefficiently allocated, being wasted either on trivial prompts that yield diminishing returns or on excessively challenging examples where the model shows negligible progress.

Recent works have investigated progressive training curricula that partition the process into several hand-crafted stages of increasing difficulty (Luo et al., 2025; Song et al., 2025). However, such partitioning is overly coarse and struggles to align with the evolving capabilities of reasoning models during training. Other approaches apply online data filtering by generating and then pruning samples (Yu et al., 2025; Bae et al., 2025; Lin et al., 2025). Yet, this paradigm does little to conserve

---

[*]Equal Contribution.
[†]Corresponding authors.

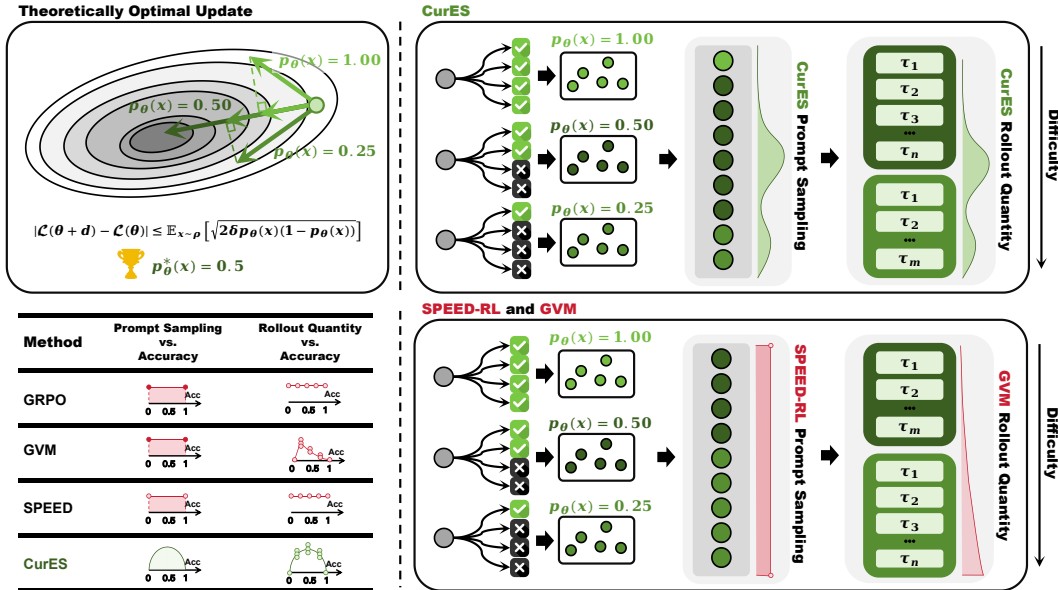

Figure 1: Illustration of our theoretical and practical contributions. The first part presents our theoretical analysis, which establishes the relationship between the gradient efficiency and models' question-answering accuracy, denoted as $p_\theta(x)$. Building upon these insights, we develop CurES, a practical method that initially estimates $p_\theta(x)$ using a small rollout quantity, then reallocates prompt sampling probabilities and rollout quantities based on the estimated accuracy. We progressively enhance the confidence of these accuracy estimates through posterior estimation. The figure further contrasts CurES with existing approaches, highlighting differences in managing prompt sampling distributions of Speed-RL (Zhang et al., 2025) and rollout quantities of GVM (Yao et al., 2025). computational resources and instead leads to suboptimal sample efficiency. Additional studies have begun exploring dynamic computation reallocation across prompts with minimal overhead (Yao et al., 2025; Zhang et al., 2025; Shi et al., 2025). Nevertheless, these techniques address only isolated facets of training acceleration, without fully accounting for the problem's inherent dynamism.

In this work, we first analyze the efficiency of training optimization for reasoning models from the perspective of gradients, elucidating its close relationship with the sampling probability distribution of prompts and the allocation of rollout quantities across these prompts. Our analysis reveals that the prompt sampling distribution directly influences the speed of gradient descent, while the allocation of rollout quantities affects the consistency and stability of overall gradient updates. Leveraging these insights, we propose CurES, a practical training method. CurES first estimates prompt difficulty via models' question-answering accuracy, then reallocates prompt sampling probabilities and rollout quantities accordingly. During training, the confidence in these accuracy estimates is progressively refined through posterior estimation based on previously sampled data, thereby improving the robustness of the allocation process. Figure 1 illustrates the overall approach, and our contributions are summarized below:

- We provide a theoretical analysis from the gradient perspective, elucidating the intrinsic relationship between training optimization efficiency and prompt sampling distribution, as well as the allocation of rollout quantities across prompts.
- Guided by the theoretical analysis, we propose a practical training method that integrates Bayesian posterior estimation, achieving enhanced efficiency and stability in reasoning model training with minimal computational overhead.
- Experimental results show that our approach achieves state-of-the-art performance across diverse mathematical reasoning datasets and model scales, accelerating convergence by up to 5.5 times compared to uniform sampling baselines like GRPO.

## 2 RELATED WORKS

**Gradient Analysis in Optimization.** Gradient analysis plays a pivotal role in understanding and improving optimization processes (Ruder, 2016; Bottou et al., 2018; Yang et al., 2024). A com-

mon application involves leveraging gradient analysis to reduce variance in iterative optimization and enhance training stability (Medyakov et al., 2025; Yuan et al., 2024). Additionally, it facilitates adaptive learning rate adjustments by dynamically scaling updates based on gradient histories (Dereich et al., 2024; Chen et al., 2024). Gradient analysis also guides data selection and curriculum learning strategies, enabling dynamic adjustment of training data difficulty (Yao et al., 2025; Li et al., 2024b). Furthermore, it aids in detecting anomalous samples to improve data quality by identifying outliers in the gradient space (Chhabra et al., 2024). As gradients are directly tied to model optimization, they provide the most immediate insights into training dynamics. Theoretical analysis of gradients enables predictions of convergence rates and bounds, ensuring robust optimization guarantees (Zhao & Xu, 2024). In this work, we examine the interplay between model optimization and sample selection from a gradient perspective, deriving methods to enhance training efficiency.

**Curriculum Learning and Data Selection in RLVR.** Effective data selection is critical for optimizing RLVR training, yet designing curricula that align with the dynamic capabilities of LLMs remains challenging. Progressive training curricula, such as those proposed in (Luo et al., 2025; Song et al., 2025), partition training into hand-crafted stages of increasing difficulty. However, these static approaches often fail to adapt to the evolving proficiency of models during training. Self-evolving curriculum learning (Chen et al., 2025) addresses this by selecting sub-datasets of appropriate difficulty based on the model's current performance. Nevertheless, such approaches still operate at the dataset granularity, resulting in a relatively coarse-grained curriculum mechanism. Online data filtering methods, such as those in (Yu et al., 2025; Bae et al., 2025; Lin et al., 2025; Xu et al., 2025), generate and prune samples to focus on high-impact data, yet since they still require executing the overall rollout generation phase, they fail to effectively reduce rollout-induced computational overhead, resulting in wasted sampling budget. Recent efforts have explored dynamic computation reallocation to prioritize prompts with higher training utility (Yao et al., 2025; Zhang et al., 2025; Shi et al., 2025). However, these methods address only specific aspects of training acceleration. Moreover, these dynamic computation allocation approaches often require explicit estimation of dataset-related parameters such as prompt difficulty, which could introduce additional and potentially unnecessary overhead. One direction to mitigate this is to incorporate Bayesian posterior estimation techniques (Qu et al., 2025a; Wang et al., 2025; Qu et al., 2025b) to guide prompt selection or active sampling, thereby reducing computational cost. Our work proposes CurES, which first systematically analyzes the correlation between gradient efficiency and problem difficulty from a gradient perspective. This analysis then informs our curriculum learning strategy, enabling the dynamic adjustment of prompt sampling probabilities and rollout quantities. Concurrently, to minimize computational overhead, we integrate Bayesian posterior estimation to optimize parameter estimation cost, resulting in improved overall training efficiency with minimal computational burden.

## 3 PRELIMINARIES

Reinforcement Learning with Verifiable Reward (RLVR) represents a specialized reinforcement learning paradigm tailored for reasoning tasks, where reward signals can be deterministically verified through programmatic means. This approach is particularly well-suited for domains such as mathematical reasoning, code generation, and logical deduction, where correctness criteria are objectively defined (Lambert et al., 2024; Guo et al., 2025; Team et al., 2025). Formally, given a policy model $\pi_\theta$ and a prompt distribution $\rho$, the RLVR objective aims to maximize the expected reward while constraining policy updates within a trust region:

$$\mathcal{L}(\theta) = -\mathbb{E}_{x \sim \rho, y \sim \pi_\theta(\cdot|x)}\big[A_{\theta_{\text{old}}}(x, y)\big], \quad \text{subject to } \mathbb{E}_{x \sim \mathcal{D}}[D_{\text{KL}}(\pi_{\theta_{\text{old}}}(\cdot|x)\|\pi_\theta(\cdot|x))] \le \delta. \quad (1)$$

Here, $A_{\theta_{\text{old}}}(x, y) = r(x, y) - \mathbb{E}_{y \sim \pi_{\theta_{\text{old}}}}[r(x, y)]$ denotes the advantage function, $\theta_{\text{old}}$ represents the policy parameters from the previous iteration, and $\delta$ defines the trust region boundary that prevents excessive policy divergence.

A key characteristic of RLVR is its reward formulation. Unlike preference-based RLHF that relies on subjective human judgments, RLVR employs a verifiable reward function defined as:

$$r(x, y) = \begin{cases} 1, & \text{if } y \text{ is the correct answer for } x \\ 0, & \text{otherwise} \end{cases} \quad (2)$$

The straightforward reward function design partially mitigates the issue of reward hacking.

# 4 METHODOLOGY

In this section, we introduce CurES, a novel method designed to enhance the training efficiency of Reasoning LLMs. We begin by establishing a theoretical connection between gradient optimization efficiency and two key factors: the sampling distribution of prompts and the allocation of rollout quantities across these prompts. Based on this analysis, CurES first leverages the estimation of the model's question-answering accuracy to assess prompt difficulty, which is then used to guide an optimal sampling strategy and rollout quantity allocation. By leveraging Bayesian posterior estimation, we progressively refine the confidence in these accuracy estimates using historical sampling data, ensuring robust and adaptive resource allocation with minimal computational overhead.

## 4.1 PROMPT DIFFICULTY CAPS OPTIMIZATION POTENTIAL

To facilitate subsequent derivations, we define the question difficulty as the model's accuracy in answering the question. Given a policy model $\pi_\theta$ and a binary reward function $r(x, y)$ in Eq. (2), the expression for the model's question-answering accuracy $p_\theta$ is given by:

$$p_\theta(x) = \mathbb{E}_{y \sim \pi_\theta}[r(x, y)]. \tag{3}$$

To investigate how prompt difficulty influences model gradient updates, we first consider the following optimization problem for a given prompt $x$:

$$\min \mathcal{L}(x; \theta) = \min -\mathbb{E}_{y \sim \pi_\theta(\cdot|x)}\big[A_{\theta_{\text{old}}}(x, y)\big], \quad \text{s.t. } D_{\text{KL}}(\pi_{\theta_{\text{old}}}(\cdot|x) \| \pi_\theta(\cdot|x)) \leq \delta. \tag{4}$$

Here, we separately analyze the impact of different prompts on the loss function and theoretically examine how varying prompt difficulty levels affect model training efficiency.

We employ the Lagrange multiplier method to solve the above problem. First, we set $\theta = \theta_{\text{old}} + d$ and reformulate the problem as follows:

$$d^* = \underset{d}{\text{argmin}} \ \mathcal{L}(x; \theta_{\text{old}} + d) + \lambda(D_{\text{KL}}(\pi_{\theta_{\text{old}}}(\cdot|x) \| \pi_{\theta_{\text{old}}+d}(\cdot|x)) - \delta). \tag{5}$$

By performing first-order Taylor expansion on the loss function $\mathcal{L}(x; \theta)$ and second-order Taylor expansion on the KL divergence term $D_{\text{KL}}(\pi_{\theta_{\text{old}}}(\cdot|x) \| \pi_\theta(\cdot|x))$, followed by simplification, we derive the following equation:

$$d^* = \underset{d}{\text{argmin}} \ \mathcal{L}(x; \theta_{\text{old}}) + \nabla_\theta \mathcal{L}(x; \theta)^\mathsf{T}\Big|_{\theta=\theta_{\text{old}}} d + \frac{\lambda}{2} d^\mathsf{T} F(x; \theta_{\text{old}}) d - \lambda\delta, \tag{6}$$

where $F(x; \theta)$ is the Fisher Information Matrix, a metric quantifying the information that observed data provides about parameter estimates in probabilistic models. In reinforcement learning, it primarily serves to construct more reasonable parameter update directions, thereby improving the efficiency and stability of policy optimization. Here, $F(x; \theta)$ is represented as

$$F(x; \theta) = \mathbb{E}_{y \sim \pi_\theta} \left[ \nabla_\theta \log \pi_\theta(y|x) \nabla_\theta \log \pi_\theta(y|x)^\mathsf{T} \right]. \tag{7}$$

Through mathematical derivation to address the problem of Eq. (6), we obtain the following results:

$$d^* = -\frac{1}{\lambda} F^{-1}(x; \theta_{\text{old}}) \nabla_\theta \mathcal{L}(x; \theta)\Big|_{\theta=\theta_{\text{old}}}, \tag{8}$$

$$\lambda = \sqrt{\frac{\nabla_\theta \mathcal{L}(x; \theta)^\top \big|_{\theta=\theta_{\text{old}}} F^{-1}(x; \theta_{\text{old}}) \nabla_\theta \mathcal{L}(x; \theta) \big|_{\theta=\theta_{\text{old}}}}{2\delta}}. \tag{9}$$

With Eq. (8) and Eq. (9), we derive the expression for the update of the loss function:

$$|\mathcal{L}(x; \theta_{\text{old}} + d) - \mathcal{L}(x; \theta_{\text{old}})| = \sqrt{2\delta \nabla_\theta \mathcal{L}(x; \theta)^\mathsf{T}\Big|_{\theta=\theta_{old}} F^{-1}(x; \theta_{\text{old}}) \nabla_\theta \mathcal{L}(x; \theta)\Big|_{\theta=\theta_{\text{ old}}}}. \tag{10}$$

According to the definition, the binary reward function $r(x, y)$ serves as an unbiased estimator of the model's question-answering accuracy $p_\theta(x)$, i.e., $p_\theta(x) = \mathbb{E}_{y \sim \pi_\theta}[r(x, y)]$. Through the application of the Cramér-Rao inequality, we derive the following fundamental result:

$$|\mathcal{L}(x; \theta_{\text{old}} + d) - \mathcal{L}(x; \theta_{\text{old}})| \leq \sqrt{2\delta p_{\theta_{\text{old}}}(x)\left(1 - p_{\theta_{\text{old}}}(x)\right)}. \tag{11}$$

Therefore, for the loss function $\mathcal{L}(\theta)$, its optimization potential exhibits the following relationship with prompt difficulty:

$$|\mathcal{L}(\theta_{\text{old}} + d) - \mathcal{L}(\theta_{\text{old}})| \leq \mathbb{E}_{x \sim \rho} \left[ \sqrt{2\delta p_{\theta_{\text{old}}}(x)(1 - p_{\theta_{\text{old}}}(x))} \right]. \tag{12}$$

This demonstrates that the convergence rate of the model's loss function is intrinsically related to the difficulty of the prompt dataset, which is quantified by the model's answering accuracy. To accelerate training, the sampling distribution $\rho$ should assign varied probabilities to prompts based on difficulty while maintaining a balance with exploration. Thus, we seek the optimal sampling distribution $\rho$ under the entropy maximization constraint. Concretely, we address the following problem:

$$\max \; \mathbb{E}_{x \sim \rho} \left[ \sqrt{2\delta p_{\theta_{\text{old}}}(x)(1 - p_{\theta_{\text{old}}}(x))} + \alpha \mathcal{H}(\rho) \right], \quad \text{s.t.} \; \sum_{i=1}^{N} \rho(x_i) = 1. \tag{13}$$

Solving the aforementioned problem, we obtain the optimal sampling distribution as follows:

$$\rho^*(x) = \frac{\exp\left(\sqrt{p_{\theta_{\text{old}}}(x)(1 - p_{\theta_{\text{old}}}(x))}/\tau\right)}{\sum_{x'} \exp\left(\sqrt{p_{\theta_{\text{old}}}(x')(1 - p_{\theta_{\text{old}}}(x'))}/\tau\right)}, \tag{14}$$

where $\tau = \frac{\alpha}{\sqrt{2\delta}}$ is a hyperparameter. For the theoretical proof please refer to Appendix A.1.

## 4.2 Closing the Gap with Theoretical Bound

In the previous section, we derived an upper bound on the gradient update for a given prompt. However, due to the high computational cost of the natural gradient method, it is often avoided in practice, and the theoretical result is instead used to guide prompt sampling. During actual gradient updates, we aim to closely approximate the theoretical efficiency limit within a trust region bounded by a KL divergence constraint of $\delta$. Specifically, after sampling a batch of $m$ prompts, we seek to optimize operations to approach the bound. Within the curriculum learning framework, we consider optimizing the allocation of rollout quantities across prompts under a fixed total rollout budget of $N$ to minimize the following loss function:

$$\min \; \mathbb{E} \left[ \left( \mathcal{L}(\hat{\theta}) - \mathcal{L}(\theta_{\text{old}}) - \left( -\mathbb{E}_{x \sim \rho} \left[ \sqrt{2\delta p_{\theta_{\text{old}}}(x)(1 - p_{\theta_{\text{old}}}(x))} \right] \right) \right)^2 \right], \quad \text{s.t.} \; \sum_{i=1}^{m} n_i = N. \tag{15}$$

Here, $\hat{\theta}$ denotes the updated model parameters obtained from $\theta_{\text{old}}$ after applying the practical gradient update, i.e.:

$$\hat{\theta} = \theta_{\text{old}} - \eta \nabla_\theta \hat{\mathcal{L}}(\theta) \big|_{\theta = \theta_{\text{old}}}, \quad \hat{\mathcal{L}}(\theta) = -\frac{1}{m} \sum_{i=1}^{m} \frac{1}{n_i} \sum_{y_j \in \mathcal{D}_i} \left[ \frac{\pi_\theta(y_j | x_i)}{\pi_{\theta_{\text{old}}}(y_j | x_i)} A_{\theta_{\text{old}}} \right]. \tag{16}$$

Where $\eta$ is the learning rate and $n_i$ denotes the number of sampled rollouts for question $x_i$. We assume that $\eta$ is chosen such that the policy update remains within a KL divergence constraint of $\delta$.

For convenience, we denote $g = \nabla_\theta \mathcal{L}(\theta) \big|_{\theta = \theta_{\text{old}}}, \hat{g} = \nabla_\theta \hat{\mathcal{L}}(\theta) \big|_{\theta = \theta_{\text{old}}}$. By simplifying the loss function, we can show that the optimization problem reduces to the following:

$$\min \; g^\top \mathbb{V}(\hat{g}) g, \quad \text{s.t.} \; \sum_{i=1}^{m} n_i = N. \tag{17}$$

The theoretical gradient direction $g$ is typically unknown, and we seek to control the uncertainty of the estimator in all possible directions. Therefore, we instead minimize the total variance $\text{Tr}(\mathbb{V}(\hat{g}))$, which corresponds to uniformly reducing the variance in all directions. This approach is a widely adopted technique for variance estimation(Bottou et al., 2018; Papini et al., 2018; Wang et al., 2013). In other words, we consider the following optimization problem:

$$\min \; \text{Tr}(\mathbb{V}(\hat{g})), \quad \text{s.t.} \; \sum_{i=1}^{m} n_i = N. \tag{18}$$

By expanding the variance of the aforementioned gradient, we isolate the rollout quantities $n_i$ to facilitate analysis:

$$\min \frac{1}{m^2} \sum_{i=1}^{m} \frac{\text{Tr}\left(\mathbb{V}_{y \sim \pi_{\theta_{\text{old}}}}\left(h\left(y, x_i; \theta_{\text{old}}\right)\right)\right)}{n_i}, \quad \text{s.t.} \sum_{i=1}^{m} n_i = N. \tag{19}$$

where $h(x, y; \theta) = \frac{\nabla_\theta \pi_\theta(y_j|x_i)}{\pi_{\theta_{\text{old}}}(y_j|x_i)} A_{\theta_{\text{old}}}(x_i, y_j)$. By applying the Lagrange multiplier method to solve the above problem, we obtain the optimal solution as follows:

$$n_i = \frac{\sigma_i}{\sum_j \sigma_j} N, \quad \sigma_i = \sqrt{\text{Tr}\left(\mathbb{V}_{y \sim \pi_{\theta_{\text{old}}}}\left(h\left(y, x_i; \theta_{\text{old}}\right)\right)\right)}. \tag{20}$$

The remaining challenge is computing $\sigma_i$. By expanding the variance and noting that the advantage function can be evaluated based on whether the rollout $y$ is correct, i.e., $A_{\theta_{\text{old}}}(x, y) = \mathbb{I}(y \text{ is correct for } x) - p_{\theta_{\text{old}}}(x)$, we derive the following symmetric computational form:

$$\text{Tr}\left(\mathbb{V}_{y \sim \pi_{\theta_{\text{old}}}}\left(h\left(y, x_i; \theta_{\text{old}}\right)\right)\right)$$

$$= p_{\theta_{\text{old}}}(x_i)\left(1 - p_{\theta_{\text{old}}}(x_i)\right)^2 \mathop{\mathbb{E}}_{y \sim \pi_{\theta_{\text{old}}}, r=1}\left[\left\|\nabla_\theta \log \pi_\theta(y|x_i)|_{\theta=\theta_{\text{old}}}\right\|^2\right]$$

$$+ \left(p_{\theta_{\text{old}}}(x_i)\right)^2 \left(1 - p_{\theta_{\text{old}}}(x_i)\right) \mathop{\mathbb{E}}_{y \sim \pi_{\theta_{\text{old}}}, r=0}\left[\left\|\nabla_\theta \log \pi_\theta(y|x_i)|_{\theta=\theta_{\text{old}}}\right\|^2\right] \tag{21}$$

$$- p_{\theta_{\text{old}}}(x_i)^2 \left(1 - p_{\theta_{\text{old}}}(x_i)\right)^2 \left\|\mathop{\mathbb{E}}_{y \sim \pi_{\theta_{\text{old}}}, r=1}\left[\nabla_\theta \log \pi_\theta(y|x_i)\right)\right] - \mathop{\mathbb{E}}_{y \sim \pi_{\theta_{\text{old}}}, r=0}\left[\nabla_\theta \log \pi_\theta(y|x_i)\right]\right\|^2.$$

The optimized formula decomposes the variance estimation problem into two categories based on answer correctness, integrating it with the prompt difficulty estimation from Section 4.1. By leveraging algebraic operations on prompt difficulty and policy gradients, it reuses difficulty estimates from sampling and transforms variance estimation into a more tractable form. The theoretical proof is provided in Appendix A.2.

### 4.3 Prompt Difficulty Assessment and Algorithmic Implementation

Estimating prompt difficulty is crucial for both sampling questions and allocating rollout quantities. However, difficulty changes dynamically during policy training, making accurate estimation challenging. A straightforward approach is to add a pre-evaluation step before each sampling, but this increases computational overhead and fails to leverage new samples for posterior estimation to improve confidence. To address this, we propose a Bayesian inference framework that decomposes rollout into a multi-stage mini-batch process. This refines the posterior estimation of the dataset, dynamically adjusting the sampling distribution based on updated difficulty assessments.

Specifically, as the model $\pi_{\theta_{\text{old}}}$ rollouts on a prompt $x_i$ multiple times, the number of correct answers follows a binomial distribution with success probability $p_{\theta_{\text{old}}}(x_i)$. We can assume that $p_{\theta_{\text{old}}}(x_i)$ follows a Beta distribution, the conjugate prior of the binomial distribution, which is a widely adopted technique in Bayesian inference (Kruschke, 2010; Qu et al., 2025a):

$$p_{\theta_{\text{old}}}(x_i) \sim \text{Beta}(\alpha_0(x_i), \beta_0(x_i)), \tag{22}$$

where $\alpha_0(x_i)$ and $\beta_0(x_i)$ can be interpreted as the counts of correct and incorrect answers during sampling, which can be initialized using a small batch of sampled data for cold-start estimation.

Since the Beta distribution is conjugate to the binomial likelihood, the posterior distribution remains Beta-distributed after observing new samples. Let $\alpha_{t-1}(x_i)$ and $\beta_{t-1}(x_i)$ denote the cumulative counts of correct and incorrect answers for prompt $x_i$ up to step $t-1$. If, at step $t$, a mini-batch generates $n_i$ answer with $s$ correct, the posterior distribution for $p_{\theta_{\text{old}}}(x_i)$ after $t$ steps is:

$$\alpha_t(x_i) = \alpha_{t-1}(x_i) + s, \quad \beta_t(x_i) = \beta_{t-1}(x_i) + n_i - s, \tag{23}$$

$$p_{\theta_{\text{old}}}(x_i) \sim \text{Beta}(\alpha_t(x_i), \beta_t(x_i)). \tag{24}$$

To reduce randomness, we use the mean of the Beta distribution to estimate prompt difficulty in our experiments.

---

Algorithm 1: From Gradient Analysis to Efficient Curriculum Learning for Reasoning LLMs (CurES)

---

1: **Input**: initial policy model $\pi_\theta$; reward function $r(x, y)$; prompt dataset $\mathcal{D} = \{x_i\}_{i=1}^{|\mathcal{D}|}$; number of iterations $T$; prompt batch size $m$; learning rate $\eta$; parameter $\tau$; total rollout budget $N$; parameter estimation sample size $N'$; number of steps per iteration $M$.
2: **for** each $t \in 1, \ldots, T$ **do**
3:   ▷ *Parameter Estimation Stage*
4:   **for** each $x_i \in \mathcal{D}_t$ **do**
5:     Sample $k = N'$ rollouts $\{y_{i,1}, \ldots, y_{i,k}\} \sim \pi_{\theta_{\text{old}}}(\cdot|x_i)$.
6:     Compute rewards $r(x_i, y_{i,j})$.
7:     Initialize counts and difficulty estimations $p(x_i)$ according to Eq. (22).
8:     Initialize sampling probabilities $\rho^*$ according to Eq. (14).
9:     Compute gradient contribution according to Eq. (21).
10:   **end for**
11:   ▷ *Large-scale Training Stage*
12:   **for** step $= 1, \cdots, M$ **do**
13:     Update the old policy model $\pi_{\theta_{\text{old}}} \leftarrow \pi_\theta$.
14:     Sample a batch of prompts $\mathcal{B} = \{x_i\}_{i=1}^m$ with replacement according to $\rho^*$.
15:     Obtain rollout quantities $n_i$ for $x_i \in \mathcal{B}$ according to Eq. (20).
16:     **for** each $x_i \in \mathcal{B}$ **do**
17:       Sample $k = n_i$ rollouts $\{y_{i,1}, \ldots, y_{i,k}\} \sim \pi_{\theta_{\text{old}}}(\cdot|x_i)$.
18:       Compute rewards $r(x_i, y_{i,j})$.
19:       Update counts and difficulty estimations $p(x_i)$ according to Eq. (23).
20:     **end for**
21:     Update sampling probabilities $\rho^*$ according to Eq. (14).
22:     Update policy $\pi_\theta$ by applying RL training.
23:   **end for**
24: **end for**
25: **Return** $\pi_\theta$.

---

This estimation approach enables modeling and estimating the difficulty of each prompt with minimal overhead. However, as the model's performance evolves during training, the estimation process is susceptible to distribution shift, which becomes more pronounced with increasing training steps. To mitigate this issue, we adopt a straightforward solution inspired by GVM. Specifically, we divide the dataset into $T$ non-overlapped subsets and perform iterative training on these subsets. We train the model for a fixed training steps of $M$ in every iteration. The estimations of prompt difficulty and gradient variance are reset when a new iteration begins. This method effectively alleviates distribution shift without introducing significant computational overhead. Moreover, the iterative process allows the model to adaptively adjust its sampling allocation based on its own evolving capabilities throughout training. For further details, please refer to Algorithm 1. In each iteration of training, our algorithm operates in two phases. The first phase performs a lightweight parameter estimation stage, where for each prompt $x_i$, we draw a small sample of $N'$ rollouts. This enables us to obtain preliminary estimates of its prompt difficulty and corresponding gradient variance. The second phase is the large-scale training stage, where each step we sample a batch of $m$ prompts and collect a total of $N$ rollouts, corresponding to an average of $n = N/m$ rollouts per prompt.

## 5 EXPERIMENTS

In this section, we present comprehensive experimental results and analysis of our CurES with other baselines. Our experiments focus on the following research questions:

- **RQ1:** Does CurES outperform other related baseline methods across various benchmarks?
- **RQ2:** How does CurES adapt its sampling strategy to accelerate learning efficiency?
- **RQ3:** Does CurES enhance sampling efficiency compared to other baseline methods?

**Training Details.** We employ VERL (Sheng et al., 2025) as our training framework and initialize our policy using Qwen2.5-Math models (1.5B and 7B parameters). For the training dataset, we utilize

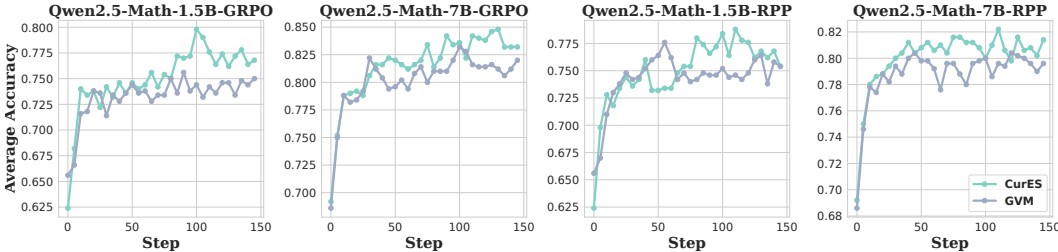

Figure 2: Comparison of learning curves between CurES and GVM across different backbone models. CurES consistently outperforms GVM under the same number of training steps, demonstrating more efficient utilization of samples.

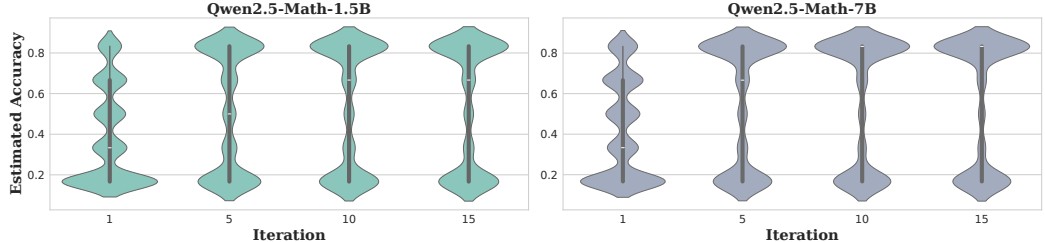

Figure 3: The evolution of the estimated accuracy distributions for the Qwen2.5-Math-1.5B (left) and 7B (right) models across 15 iterations. Each violin shows the distribution of accuracy across samples: the width reflects density, the central line marks the median.

Numina-Math (Li et al., 2024a), partitioning it into 15 subsets following GVM (Yao et al., 2025). We conduct iterative training across these subsets, resulting in 15 training iterations. At the beginning of each iteration, we perform 4 rollouts per prompt to establish an initial difficulty distribution and an assignment of rollout quantities under a total sample budget of $8 \times 1024$. During training, we sample prompts according to the difficulty distribution with replacement and conduct rollouts according to the assigned rollout quantities. To make a fair comparison with GVM, we train 10 steps in each iteration. We employ GRPO (Shao et al., 2024) and REINFORCE++ (RPP) (Hu et al., 2025) as advantage estimators for all methods. The learning rate is set to a constant $1 \times 10^{-6}$.

**Evaluation Benchmarks.** To evaluate the complex reasoning capabilities, we choose a broad set of challenging reasoning benchmarks, including MATH500 (Hendrycks et al., 2021), AIME 2024 and 2025 (Li et al., 2024a), AMC 2023 (Li et al., 2024a), GSM8K (Cobbe et al., 2021), Gaokao-EN 2023 (Zhang et al., 2023), Mineva (Lewkowycz et al., 2022) and OlympiadBench (He et al., 2024). These benchmarks comprehensively evaluate mathematical reasoning capabilities. Since AIME 2024, 2025 and AMC 2023 are highly challenging competition benchmarks, which are of limited sizes of test samples, we present the results averaged over 16 runs.

**Baselines.** To demonstrate the reasoning ability of our CurES, we compare it with many strong baseline methods: GRPO (Shao et al., 2024), RPP (Hu et al., 2025), Speed-RL (Zhang et al., 2025) and GVM (Yao et al., 2025). Specifically, GRPO and RPP are commonly used in training mathematical problem solving models. Speed-RL is an adaptive online RL curriculum that selectively chooses samples of intermediate difficulty to maximize learning efficiency (i.e., samples whose accuracy is not $0$ or $1$). GVM is a prompt-specific dynamic sample allocation strategy designed to minimize stochastic gradient variance under a computational budget constraint.

## 5.1 OVERALL PERFORMANCE (RQ1)

We present the learning curve of Qwen2.5-Math-1.5B and 7B models trained with different methods in Figure 2. Across all configurations, CurES exhibits higher progressive and final accuracy compared to GVM. This advantage originates from two key differences: (i) CurES adaptively allocates prompt sampling probabilities based on estimated success rates, which our theoretical analysis confirms enhances training efficiency; (ii) while GVM monotonically decreases rollout allocation as accuracy increases, CurES allocates more rollout budget to prompts of moderate difficulty in Figure 4, resulting in more consistent training gradients and improved training stability.

Table 1: Quantitative results of different methods across various datasets. The best and second best results are in **bold** and underlined.

| Method | Pass@1 | | | | | Average@16 | | | Avg. |
|---|---|---|---|---|---|---|---|---|---|
| | MATH500 | GSM8K | GAO23 | MINERVA | OLYM | AIME24 | AIME25 | AMC23 | |
| Qwen2.5-Math-1.5B | 40.20 | 43.90 | 25.19 | 11.40 | 21.04 | 1.67 | 1.67 | 14.84 | 20.00 |
| +GRPO | 73.80 | **86.43** | 48.83 | 27.94 | 35.41 | 8.54 | 6.67 | 45.47 | 41.64 |
| +RPP | 64.80 | 82.94 | 42.08 | 21.32 | 29.19 | 4.17 | 3.33 | 39.06 | 35.86 |
| +Speed-RL-GRPO | 68.80 | 85.67 | 47.14 | 27.40 | 35.56 | 12.08 | 6.88 | 47.19 | 41.34 |
| +Speed-RL-RPP | 65.80 | 85.67 | 48.31 | 27.94 | 36.30 | 12.58 | 10.08 | 47.34 | 41.75 |
| +GVM-GRPO | 74.80 | 84.23 | 48.83 | 27.21 | 35.56 | 10.21 | 11.25 | 50.47 | 42.82 |
| +GVM-RPP | 75.40 | 84.00 | 49.61 | 24.63 | 35.56 | 11.46 | 6.04 | 50.94 | 42.21 |
| +CurES-GRPO | **77.20** | 85.97 | **51.43** | **31.62** | **37.33** | **13.33** | 10.42 | **52.19** | **44.94** |
| +CurES-RPP | 75.40 | 85.82 | **51.43** | 28.31 | 37.04 | 12.71 | **11.46** | 50.94 | 44.14 |
| Qwen2.5-Math-7B | 60.20 | 72.40 | 44.68 | 22.79 | 30.81 | 7.92 | 1.88 | 27.19 | 33.48 |
| +GRPO | 80.00 | 91.43 | 51.43 | 31.99 | 38.37 | 20.00 | 10.00 | 57.50 | 47.59 |
| +RPP | 81.20 | 91.89 | 55.58 | **39.71** | 40.00 | 18.54 | 11.67 | 62.81 | 50.18 |
| +Speed-RL-GRPO | 82.80 | 88.70 | 55.58 | 29.41 | 42.37 | 20.21 | 11.46 | 60.16 | 48.84 |
| +Speed-RL-RPP | 78.60 | 91.81 | 53.77 | 37.13 | 42.96 | 17.29 | 12.08 | 62.81 | 49.56 |
| +GVM-GRPO | 81.60 | 91.28 | 54.03 | 32.72 | 42.67 | 23.54 | **15.00** | 64.31 | 50.64 |
| +GVM-RPP | 81.60 | 90.07 | 55.32 | 29.04 | 40.30 | 17.50 | 8.33 | 53.44 | 46.95 |
| +CurES-GRPO | **84.80** | **92.27** | **56.62** | 37.87 | **43.56** | **24.58** | 15.21 | **64.38** | **52.41** |
| +CurES-RPP | 81.80 | 91.89 | 54.55 | 33.09 | 40.59 | 23.33 | 12.92 | 58.75 | 49.62 |

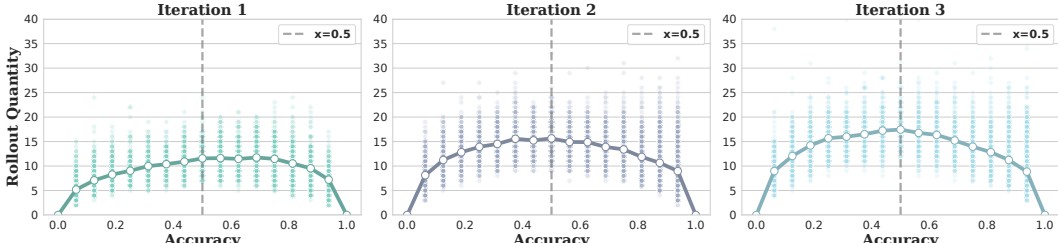

Figure 4: Allocation of rollout quantities with respect to accuracy in CurES at different training iterations. CurES concentrates more rollouts on moderately difficult prompts.

To demonstrate the effectiveness of our CurES, we compare it with representative baselines that are trained for the same number of steps. The main results are demonstrated in Table 1. The results clearly demonstrate that CurES consistently outperforms GVM and other baselines with both GRPO and RPP as advantage estimators. Across both model scales, CurES establishes state-of-the-art results on several datasets and consistently matches or surpasses the strongest baselines across all settings, confirming the superior generalization ability of our CurES.

## 5.2 SAMPLING BEHAVIOR (RQ2)

Figure 3 illustrates the evolution of the difficulty distribution for both the Qwen2.5-Math-1.5B and 7B models throughout the training process. At iteration 1, the estimated accuracy is broadly distributed. As training progresses, this distribution shifts toward higher values and becomes more concentrated, indicating that the models are effectively learning and mastering the presented samples. This shift also underscores the importance of redistributing prompt sampling probabilities, as the models' success rates on problems are primarily bimodal, concentrated at high and low values. Such redistribution enhances training efficiency, whereas uniform sampling followed by answer generation and accuracy-based filtering significantly reduces efficiency under this bimodal distribution.

Meanwhile, another view of how the CurES method dynamically adjusts the rollout quantities assignment during training is presented in Figure 4. The trend lines of all iterations approximate a "bell-shaped" distribution, with prompts of intermediate accuracy allocated more rollout quantities, as anticipated given their high efficiency. Furthermore, as the training procedure continues, the distribution becomes progressively sharper and narrower, indicating that CurES dynamically increases rollout quantities for moderately difficult prompts. This pattern aligns with the observation in Figure 3, which shows a gradual reduction in moderately difficult prompts as the model improves. By adaptively increasing rollout quantities, CurES compensates for their diminishing presence, ensur-

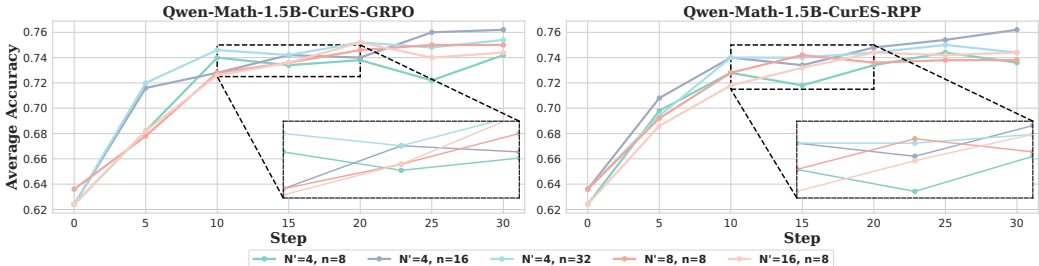

Figure 5: Performance convergence of CurES on MATH500 with different sampling configurations.

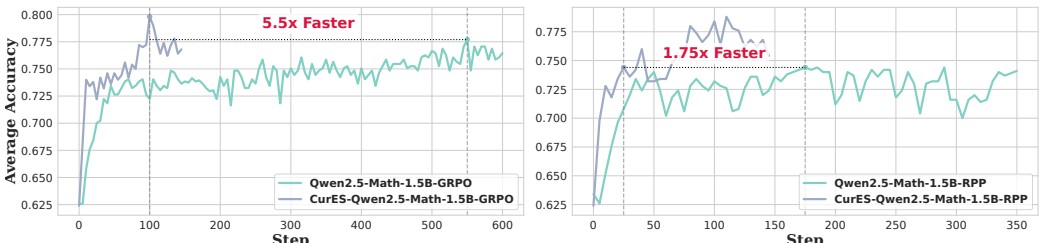

Figure 6: Efficiency comparison of CurES against baselines on MATH500. The gray dashed lines mark the number of steps CurES and each baseline required to reach the highest average accuracy within the total training duration.

ing they remain a substantial portion of each training batch. By coupling difficulty-based prompt sampling with the adaptive sample sizing, CurES sustains an abundance of informative prompts and thereby maximizes performance gains per step.

### 5.3 EFFICIENCY ANALYSIS (RQ3)

To analyze the effect of different combinations of parameter estimation sample size ($N'$) and training-phase sample budgets coefficient ($n$), indicating a training-phase sample budget of $N = m \times n$ on the convergence of model performance, we conduct a series of experiments, as shown in Figure 5. A larger $N'$ leads to a more accurate initial accuracy estimation, while a larger $n$ provides a greater computation budget. The results show that increasing either $N'$ or $n$ does not yield a proportional performance benefit relative to the increased computational cost. This finding confirms that the core mechanism of CurES efficiently guides the model toward high-yield learning samples without the need for extensive computational overhead. In practice, relatively small values of $N'$ and $n$ are already sufficient for effective training, highlighting the high sample efficiency of our method.

We also provide a direct comparison of CurES against GRPO and RPP in Figure 6. The plots show the learning curve of each method over training steps. CurES-GRPO achieves the same peak performance as the GRPO in just $5.5\times$ fewer steps. Similarly, CurES-RPP reaches its peak performance $1.75\times$ faster than the RPP baseline. The remarkable sample efficiency is a direct consequence of CurES's ability to consistently provide the model with optimally challenging samples.

### 6 CONCLUSION

In this paper, we propose CurES, a theoretically grounded curriculum learning algorithm for RLVR. By linking gradient efficiency to accuracy, our approach adaptively prioritizes training prompts of optimal difficulty and dynamically allocates rollout budgets. Beyond the theoretical analysis, our algorithmic design leverages a Bayesian framework to track prompt difficulty in a lightweight yet adaptive manner. Specifically, we model the success rate of each prompt instance with a Beta distribution, which naturally incorporates prior information and posterior updates as new rollouts are observed. Combined with the derived sampling distribution and variance-based rollout quantity allocation, this Bayesian mechanism ensures that both question selection and sample budgeting adapt dynamically to the evolving policy, thereby maximizing training efficiency in practice. Experiments on a wide range of mathematical reasoning benchmarks show that CurES consistently outperforms strong baselines in both accuracy and convergence speed, demonstrating superior sample efficiency.

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

# A  MATHEMATICAL DERIVATIONS

**Lemma 1 (Cramér-Rao Inequality)** *Let $\{p_\theta(x), \theta \in \Theta\}$ be a Cramér-Rao regular family with parameter space $\Theta \subset \mathbb{R}^k$, where the Fisher information matrix $I(\theta)$ is non-singular. Let $g(\theta) = (g_1(\theta), \cdots, g_s(\theta))^\top$ for $s \leq k$, and assume the partial derivatives $\partial g_i(\theta)/\partial \theta_j$ exist for all $i = 1, \cdots, s$ and $j = 1, \cdots, k$. Suppose $T(X)$ is an unbiased estimator of $g(\theta)$ with finite second moment. Denote $G(\theta) = \nabla_\theta g(\theta)$, then we have*

$$\mathbb{V}_\theta(T(X)) \geq G(\theta)I^{-1}(\theta)G^\top(\theta). \tag{25}$$

## A.1  PROMPT DIFFICULTY CAPS OPTIMIZATION POTENTIAL

Given an individual sample $x$, we first consider the optimization problem as follows:

$$\begin{aligned}
\min \mathcal{L}(x; \theta) &= \min -\mathbb{E}_{y \sim \pi_\theta(\cdot|x)}\big[A_{\theta_{\mathrm{old}}}(x, y)\big], \\
&\text{s.t. } D_{\mathrm{KL}}(\pi_{\theta_{\mathrm{old}}}(\cdot|x)\|\pi_\theta(\cdot|x)) \leq \delta
\end{aligned} \tag{26}$$

We define $\theta = \theta_{\mathrm{old}} + d$ and rewrite this constrained optimization problem via the Lagrange multiplier method:

$$d^* = \underset{d}{\operatorname{argmin}} \, \mathcal{L}(x; \theta_{\mathrm{old}} + d) + \lambda(D_{\mathrm{KL}}(\pi_{\theta_{\mathrm{old}}}(\cdot|x)\|\pi_{\theta_{\mathrm{old}}+d}(\cdot|x)) - \delta). \tag{27}$$

Using the Taylor expansion formula, we have:

$$\begin{aligned}
d^* &= \underset{d}{\operatorname{argmin}} \, \mathcal{L}(x; \theta_{\mathrm{old}} + d) + \lambda(D_{\mathrm{KL}}(\pi_{\theta_{\mathrm{old}}}(\cdot|x)\|\pi_{\theta_{\mathrm{old}}+d}(\cdot|x)) - \delta) \\
&= \underset{d}{\operatorname{argmin}} \, \mathcal{L}(x; \theta_{\mathrm{old}}) - \lambda\delta \\
&\quad + \left(\nabla_\theta \mathcal{L}(x; \theta)^\top d + \lambda\nabla_\theta D_{\mathrm{KL}}(\pi_{\theta_{\mathrm{old}}}(\cdot|x)\|\pi_\theta(\cdot|x))d + \frac{\lambda}{2}d^\top\nabla_\theta^2 D_{\mathrm{KL}}(\pi_{\theta_{\mathrm{old}}}(\cdot|x)\|\pi_\theta(\cdot|x))d\right)\Big|_{\theta=\theta_{\mathrm{old}}}.
\end{aligned} \tag{28}$$

We first compute the first-order and second-order derivatives of the KL divergence term:

$$\begin{aligned}
&\nabla_\theta D_{\mathrm{KL}}(\pi_{\theta_{\mathrm{old}}}(\cdot|x)\|\pi_\theta(\cdot|x))\big|_{\theta=\theta_{\mathrm{old}}} \\
&= \nabla_\theta \mathbb{E}_{y \sim \pi_{\theta_{\mathrm{old}}}}\left[\log \pi_{\theta_{\mathrm{old}}}(y|x)\right]\big|_{\theta=\theta_{\mathrm{old}}} - \nabla_\theta \mathbb{E}_{y \sim \pi_{\theta_{\mathrm{old}}}}\left[\log \pi_\theta(y|x)\right]\big|_{\theta=\theta_{\mathrm{old}}} \\
&= -\mathbb{E}_{y \sim \pi_{\theta_{\mathrm{old}}}}\left[\nabla_\theta \log \pi_\theta(y|x)\right]\big|_{\theta=\theta_{\mathrm{old}}} \\
&= -\mathbb{E}_{y \sim \pi_{\theta_{\mathrm{old}}}}\left[\frac{\nabla_\theta \pi_\theta(y|x)}{\pi_\theta(y|x)}\right]\Big|_{\theta=\theta_{\mathrm{old}}} \\
&= \sum_y \nabla_\theta \pi_\theta(y|x)\big|_{\theta=\theta_{\mathrm{old}}} \\
&= \nabla_\theta \sum_y \pi_\theta(y|x)\big|_{\theta=\theta_{\mathrm{old}}} \\
&= 0.
\end{aligned} \tag{29}$$

$$\nabla_\theta^2 D_{\mathrm{KL}}(\pi_{\theta_{\mathrm{old}}}(\cdot|x)\|\pi_\theta(\cdot|x))\big|_{\theta=\theta_{\mathrm{old}}}$$

$$= -\nabla_\theta^2 \mathbb{E}_{y\sim\pi_{\theta_{\mathrm{old}}}}\left[\log\pi_\theta(y|x)\right]\big|_{\theta=\theta_{\mathrm{old}}}$$

$$= -\mathbb{E}_{y\sim\pi_{\theta_{\mathrm{old}}}}\left[\nabla_\theta^2\log\pi_\theta(y|x)\right]\big|_{\theta=\theta_{\mathrm{old}}}$$

$$= -\mathbb{E}_{y\sim\pi_{\theta_{\mathrm{old}}}}\left[\nabla_\theta\left(\frac{\nabla_\theta\pi_\theta(y|x)}{\pi_\theta(y|x)}\right)\right]\bigg|_{\theta=\theta_{\mathrm{old}}}$$

$$= -\mathbb{E}_{y\sim\pi_{\theta_{\mathrm{old}}}}\left[\frac{(\nabla_\theta^2\pi_\theta(y|x))\pi_\theta(y|x) - \nabla_\theta\pi_\theta(y|x))\nabla_\theta^\top\pi_\theta(y|x)}{\pi_\theta^2(y|x)}\right]\bigg|_{\theta=\theta_{\mathrm{old}}} \tag{30}$$

$$= -\mathbb{E}_{y\sim\pi_{\theta_{\mathrm{old}}}}\left[\frac{(\nabla_\theta^2\pi_\theta(y|x))|_{\theta=\theta_{\mathrm{old}}}}{\pi_{\theta_{\mathrm{old}}}(y|x)}\right] + \mathbb{E}_{y\sim\pi_{\theta_{\mathrm{old}}}}\left[\left(\frac{\nabla_\theta\pi_\theta(y|x)}{\pi_\theta(y|x)}\right)\left(\frac{\nabla_\theta\pi_\theta(y|x)}{\pi_\theta(y|x)}\right)^\top\right]\bigg|_{\theta=\theta_{\mathrm{old}}}$$

$$= \mathbb{E}_{y\sim\pi_{\theta_{\mathrm{old}}}}\left[\nabla_\theta\log\pi_\theta(y|x)\nabla_\theta\log\pi_\theta(y|x)^\top\right]\big|_{\theta=\theta_{\mathrm{old}}}$$

$$= F(x;\theta_{\mathrm{old}}).$$

where $F(x;\theta) = \mathbb{E}_{y\sim\pi_\theta}\left[\nabla_\theta\log\pi_\theta(y|x)\nabla_\theta\log\pi_\theta(y|x)^\top\right]$ is termed the Fisher information matrix. Therefore,

$$d^* = \operatorname*{argmin}_d \mathcal{L}(x;\theta_{\mathrm{old}}) + \nabla_\theta\mathcal{L}(x;\theta)^\top\big|_{\theta=\theta_{\mathrm{old}}}d + \frac{\lambda}{2}d^\top F(x;\theta_{\mathrm{old}})d - \lambda\delta. \tag{31}$$

To find the minimum, we take the derivative of the right-hand side and set it to zero:

$$\nabla_\theta\mathcal{L}(x;\theta)\big|_{\theta=\theta_{\mathrm{old}}} + \lambda F(x;\theta_{\mathrm{old}})d = 0. \tag{32}$$

So we have

$$d = -\frac{1}{\lambda}F^{-1}(x;\theta_{\mathrm{old}})\nabla_\theta\mathcal{L}(x;\theta)\big|_{\theta=\theta_{\mathrm{old}}}. \tag{33}$$

We now derive the critical point of the constraint condition:

$$\frac{1}{2}d^\top F(x;\theta_{\mathrm{old}})d \approx D_{\mathrm{KL}}(\pi_{\theta_{\mathrm{old}}}(\cdot|x)\|\pi_{\theta_{\mathrm{old}}+d}(\cdot|x)) = \delta. \tag{34}$$

By simplifying, we obtain

$$\frac{1}{2}\left(\frac{1}{\lambda^2}\nabla_\theta\mathcal{L}(x;\theta)^\top\big|_{\theta=\theta_{\mathrm{old}}}F^{-1}(x;\theta_{\mathrm{old}})\nabla_\theta\mathcal{L}(x;\theta)\big|_{\theta=\theta_{\mathrm{old}}}\right) = \delta. \tag{35}$$

Therefore, we have obtained the critical value of $\lambda$:

$$\lambda = \sqrt{\frac{\nabla_\theta\mathcal{L}(x;\theta)^\top\big|_{\theta=\theta_{\mathrm{old}}}F^{-1}(x;\theta_{\mathrm{old}})\nabla_\theta\mathcal{L}(x;\theta)\big|_{\theta=\theta_{\mathrm{old}}}}{2\delta}}. \tag{36}$$

In this case, the change of the loss function is computed as

$$\mathcal{L}(x;\theta_{\mathrm{old}}+d) - \mathcal{L}(x;\theta_{\mathrm{old}}) = \nabla_\theta\mathcal{L}(x;\theta)^\top\big|_{\theta=\theta_{\mathrm{old}}}d$$

$$= -\frac{1}{\lambda}\nabla_\theta\mathcal{L}(x;\theta)^\top\big|_{\theta=\theta_{\mathrm{old}}}F^{-1}(x;\theta_{\mathrm{old}})\nabla_\theta\mathcal{L}(x;\theta)\big|_{\theta=\theta_{\mathrm{old}}} \tag{37}$$

$$= -\sqrt{2\delta\nabla_\theta\mathcal{L}(x;\theta)^\top\big|_{\theta=\theta_{\mathrm{old}}}F^{-1}(x;\theta_{\mathrm{old}})\nabla_\theta\mathcal{L}(x;\theta)\big|_{\theta=\theta_{\mathrm{old}}}}$$

Furthermore,

$$
\begin{aligned}
\nabla_\theta \mathcal{L}(x;\theta)|_{\theta=\theta_{\text{old}}} &= \nabla_\theta - \mathbb{E}_{y\sim\pi_\theta(\cdot|x)}\big[A_{\theta_{\text{old}}}(x,y)\big]\big|_{\theta=\theta_{\text{old}}} \\
&= -\sum_y \Big[\nabla_\theta \pi_\theta(y|x)\Big(r(x,y) - \mathbb{E}_{y\sim\pi_{\theta_{\text{old}}}}[r(x,y)]\Big)\Big]\big|_{\theta=\theta_{\text{old}}} \\
&= -\sum_y \big[\nabla_\theta \pi_\theta(y|x)\,(r(x,y))\big]\big|_{\theta=\theta_{\text{old}}} \\
&= -\nabla_\theta \mathbb{E}_{y\sim\pi_\theta}[r(x,y)]\big|_{\theta=\theta_{\text{old}}} \\
&= -\nabla_\theta p_\theta(x)|_{\theta=\theta_{\text{old}}},
\end{aligned}
\tag{38}
$$

where $p_\theta(x)$ is the model's question-answering accuracy.

Consider that $r(x,y)$ is an unbiased estimator of $p_\theta(x)$, according to the Cramér-Rao inequality, we obtain:

$$
\begin{aligned}
|(\mathcal{L}(\theta_{\text{old}}+d) - \mathcal{L}(\theta_{\text{old}}))| &= |\mathbb{E}_{x\sim\rho}[\mathcal{L}(x;\theta_{\text{old}}+d) - \mathcal{L}(x;\theta_{\text{old}})]| \\
&\leq \mathbb{E}_{x\sim\rho}\big[|(\mathcal{L}(x;\theta_{\text{old}}+d) - \mathcal{L}(x;\theta_{\text{old}}))|\big] \\
&= \mathbb{E}_{x\sim\rho}\Big[\sqrt{2\delta\nabla_\theta\mathcal{L}(x;\theta)^\top\big|_{\theta=\theta_{\text{old}}} F^{-1}(x;\theta_{\text{old}})\nabla_\theta\mathcal{L}(x;\theta)\big|_{\theta=\theta_{\text{old}}}}\Big] \\
&\leq \mathbb{E}_{x\sim\rho}\Big[\sqrt{2\delta\mathbb{V}_{\theta_{\text{old}}}(r(x,y))}\Big] \\
&= \mathbb{E}_{x\sim\rho}\Big[\sqrt{2\delta p_{\theta_{\text{old}}}(x)\,(1-p_{\theta_{\text{old}}}(x))}\Big]
\end{aligned}
\tag{39}
$$

This indicates that the optimization potential of the loss function is inherently related to the difficulty of the prompt itself. To balance the trade-off between exploration and exploitation, we derive the optimal sampling distribution by solving the following objective function under the maximum entropy constraint with the hyperparameter $\alpha$:

$$
\begin{aligned}
\max\ \mathbb{E}_{x\sim\rho}\Big[\sqrt{2\delta p_{\theta_{\text{old}}}(x)\,(1-p_{\theta_{\text{old}}}(x))} + \alpha\mathcal{H}(\rho)\Big], \\
\text{s.t.}\ \sum_{i=1}^N \rho(x_i) = 1,\quad \rho(x_i) \geq 0
\end{aligned}
\tag{40}
$$

To find the optimal distribution $\rho$, we employ the method of Lagrange multipliers. The objective function becomes:

$$
\max\ \mathcal{J} = \max \sum_{j=1}^N \rho(x_j)\sqrt{2\delta p_{\theta_{\text{old}}}(x_j)\,(1-p_{\theta_{\text{old}}}(x_j))} - \alpha\sum_{j=1}^N \rho(x_j)\log\rho(x_j) + \mu\left(1 - \sum_{j=1}^N \rho(x_j)\right),
\tag{41}
$$

where $\mu$ is the Lagrange multiplier associated with the normalization constraint.

Taking the partial derivative of $\mathcal{J}$ with respect to $\rho(x_j)$:

$$
\frac{\partial\mathcal{J}}{\partial\rho(x_j)} = \sqrt{2\delta p_{\theta_{\text{old}}}(x_j)\,(1-p_{\theta_{\text{old}}}(x_j))} - \alpha(\log\rho(x_j) + 1) - \mu.
\tag{42}
$$

Then set the derivative to zero:

$$
\sqrt{2\delta p_{\theta_{\text{old}}}(x_j)\,(1-p_{\theta_{\text{old}}}(x_j))} - \alpha(\log\rho(x_j) + 1) - \mu = 0.
\tag{43}
$$

Solving for $\log\rho(x_j)$:

$$
\log\rho(x_j) = \frac{\sqrt{2\delta p_{\theta_{\text{old}}}(x_j)\,(1-p_{\theta_{\text{old}}}(x_j))}}{\alpha} - 1 - \frac{\mu}{\alpha}.
\tag{44}
$$

So we have:

$$
\rho(x_j) = \exp\left(\frac{\sqrt{2\delta p_{\theta_{\text{old}}}(x_j)\,(1-p_{\theta_{\text{old}}}(x_j))}}{\alpha} - 1 - \frac{\mu}{\alpha}\right).
\tag{45}
$$

Let $c = \exp\left(-1 - \frac{\lambda}{\alpha}\right)$, then Eq. (45) becomes

$$\rho(x_j) = c \exp\left(\frac{\sqrt{2\delta p_{\theta_{\text{old}}}(x_j)\left(1 - p_{\theta_{\text{old}}}(x_j)\right)}}{\alpha}\right). \tag{46}$$

Using the constraint $\sum_{j=1}^{N} \rho(x_j) = 1$, we can obtain

$$c = \frac{1}{\sum_{j=1}^{N} \exp\left(\frac{\sqrt{2\delta p_{\theta_{\text{old}}}(x_j)\left(1 - p_{\theta_{\text{old}}}(x_j)\right)}}{\alpha}\right)}. \tag{47}$$

So the optimal distribution is

$$\rho^*(x) = \frac{\exp\left(\frac{\sqrt{2\delta p_{\theta_{\text{old}}}(x)\left(1 - p_{\theta_{\text{old}}}(x)\right)}}{\alpha}\right)}{\sum_{x'} \exp\left(\frac{\sqrt{2\delta p_{\theta_{\text{old}}}(x')\left(1 - p_{\theta_{\text{old}}}(x')\right)}}{\alpha}\right)}. \tag{48}$$

With the substitution $\tau = \frac{\alpha}{\sqrt{2\delta}}$, the distribution finally becomes:

$$\rho^*(x) = \frac{\exp\left(\sqrt{p_{\theta_{\text{old}}}(x)\left(1 - p_{\theta_{\text{old}}}(x)\right)}/\tau\right)}{\sum_{x'} \exp\left(\sqrt{p_{\theta_{\text{old}}}(x')\left(1 - p_{\theta_{\text{old}}}(x')\right)}/\tau\right)}. \tag{49}$$

## A.2   CLOSING THE GAP WITH THEORETICAL BOUND

In the previous proof, we established a lower bound for single-step gradient descent within a $\delta$-local trust region constrained by KL divergence, i.e.,

$$
\begin{aligned}
\mathcal{L}(\theta_{\text{old}} + d) - \mathcal{L}(\theta_{\text{old}}) &= \mathbb{E}_{x \sim \rho}\left[\mathcal{L}(x; \theta_{\text{old}} + d) - \mathcal{L}(x; \theta_{\text{old}})\right] \\
&= \mathbb{E}_{x \sim \rho}\left[-\sqrt{2\delta \nabla_\theta \mathcal{L}(x;\theta)^\top\big|_{\theta=\theta_{\text{old}}} F^{-1}(x;\theta_{\text{old}}) \nabla_\theta \mathcal{L}(x;\theta)\big|_{\theta=\theta_{\text{old}}}}\right] \\
&\geq -\mathbb{E}_{x \sim \rho}\left[\sqrt{2\delta \mathbb{V}_{\theta_{\text{old}}}(r(x,y))}\right] \\
&= -\mathbb{E}_{x \sim \rho}\left[\sqrt{2\delta p_{\theta_{\text{old}}}(x)\left(1 - p_{\theta_{\text{old}}}(x)\right)}\right]
\end{aligned} \tag{50}
$$

However, due to the high computational cost of the natural gradient method, it is often avoided in practice, and the theoretical result is instead used to guide prompt sampling. During actual gradient updates, we aim to closely approximate the theoretical efficiency limit within a trust region bounded by a KL divergence constraint of $\delta$. Specifically, after sampling a batch of $m$ prompts, we seek to optimize operations to approach the bound. Within the curriculum learning framework, we consider optimizing the allocation of rollouts across prompts under a fixed total rollout budget of $N$ to minimize the following loss function:

$$\min \mathbb{E}\left[\left(\mathcal{L}(\hat{\theta}) - \mathcal{L}(\theta_{\text{old}}) - \left(-\mathbb{E}_{x \sim \rho}\left[\sqrt{2\delta p_{\theta_{\text{old}}}(x)\left(1 - p_{\theta_{\text{old}}}(x)\right)}\right]\right)\right)^2\right], \quad \text{s.t.} \sum_{i=1}^{m} n_i = N. \tag{51}$$

Here, $\hat{\theta}$ denotes the updated model parameters obtained from $\theta_{\text{old}}$ after applying the practical gradient update, i.e.:

$$\hat{\theta} = \theta_{\text{old}} - \eta \nabla_\theta \hat{\mathcal{L}}(\theta)\big|_{\theta=\theta_{\text{old}}}, \quad \hat{\mathcal{L}}(\theta) = -\frac{1}{m}\sum_{i=1}^{m}\frac{1}{n_i}\sum_{y_j \in \mathcal{D}_i}\left[\frac{\pi_\theta(y_j|x_i)}{\pi_{\theta_{\text{old}}}(y_j|x_i)}A_{\theta_{\text{old}}}\right]. \tag{52}$$

Where $\eta$ is the learning rate and $n_i$ denotes the number of sampled rollouts for question $x_i$. We assume that $\eta$ is chosen such that the policy update remains within a KL divergence constraint of $\delta$.

In fact, $\hat{\mathcal{L}}(\theta)$ is an unbiased estimator of $\mathcal{L}(\theta)$, that is:

$$
\begin{aligned}
&\mathbb{E}\left[\hat{\mathcal{L}}(\theta)\right] \\
=&\mathbb{E}\left[-\frac{1}{m}\sum_{i=1}^{m}\frac{1}{n_i}\sum_{y_j\in\mathcal{D}_i}\left[\frac{\pi_\theta(y_j|x_i)}{\pi_{\theta_{\text{old}}}(y_j|x_i)}A_{\theta_{\text{old}}}(x_i,y_j)\right]\right] \\
=&-\frac{1}{m}\sum_{i=1}^{m}\frac{1}{n_i}\sum_{y_j\in\mathcal{D}_i}\mathbb{E}_{x_i\sim\rho,y_j\sim\pi_{\theta_{\text{old}}}(\cdot|x_i)}\left[\frac{\pi_\theta(y_j|x_i)}{\pi_{\theta_{\text{old}}}(y_j|x_i)}A_{\theta_{\text{old}}}(x_i,y_j)\right] \\
=&-\frac{1}{m}\sum_{i=1}^{m}\mathbb{E}_{x_i\sim\rho,y\sim\pi_{\theta_{\text{old}}}(\cdot|x_i)}\left[\frac{\pi_\theta(y|x_i)}{\pi_{\theta_{\text{old}}}(y|x_i)}A_{\theta_{\text{old}}}(x_i,y)\right] \\
=&\mathbb{E}_{x\sim\rho,y\sim\pi_{\theta_{\text{old}}}(\cdot|x)}\left[\frac{\pi_\theta(y|x)}{\pi_{\theta_{\text{old}}}(y|x)}A_{\theta_{\text{old}}}(x,y)\right] \\
=&\mathbb{E}_{x\sim\rho,y\sim\pi_{\theta(\cdot|x)}}\left[A_{\theta_{\text{old}}}(x,y)\right] \\
=&\mathcal{L}(\theta)
\end{aligned}
\tag{53}
$$

By applying the Taylor expansion, we obtain:

$$
\begin{aligned}
\mathcal{L}(\hat{\theta})-\mathcal{L}(\theta_{\text{old}})\approx&\nabla_\theta\mathcal{L}(\theta)\big|_{\theta=\theta_{\text{old}}}(\hat{\theta}-\theta_{\text{old}}) \\
=&-\eta\nabla_\theta\mathcal{L}(\theta)^\top\big|_{\theta=\theta_{\text{old}}}\nabla_\theta\hat{\mathcal{L}}(\theta)\big|_{\theta=\theta_{\text{old}}}
\end{aligned}
\tag{54}
$$

where

$$
\nabla_\theta\mathcal{L}(\theta)=-\mathbb{E}_{x\sim\rho,y\sim\pi_\theta(\cdot|x)}\left[A_{\theta_{\text{old}}}(x,y)\nabla_\theta\log\pi_\theta(y|x)\right],
\tag{55}
$$

$$
\nabla_\theta\hat{\mathcal{L}}(\theta)=-\frac{1}{m}\sum_{i=1}^{m}\frac{1}{n_i}\sum_{y_j\in\mathcal{D}_i}\left[\frac{\nabla_\theta\pi_\theta(y_j|x_i)}{\pi_{\theta_{\text{old}}}(y_j|x_i)}A_{\theta_{\text{old}}}(x_i,y_j)\right]
\tag{56}
$$

For convenience, we adopt the following notation:

$$
g=\nabla_\theta\mathcal{L}(\theta)\big|_{\theta=\theta_{\text{old}}},\quad\hat{g}=\nabla_\theta\hat{\mathcal{L}}(\theta)\big|_{\theta=\theta_{\text{old}}}
\tag{57}
$$

$$
\Delta_{\text{theo}}=\mathbb{E}_{x\sim\rho}\left[\sqrt{2\delta\cdot p_{\theta_{\text{old}}}(x)\left(1-p_{\theta_{\text{old}}}(x)\right)}\right]
\tag{58}
$$

Therefore, the original problem can then be simplified as follows:

$$
\begin{aligned}
&\mathbb{E}\left[\left(\mathcal{L}(\hat{\theta})-\mathcal{L}(\theta_{\text{old}})-\left(-\mathbb{E}_{x\sim\rho}\left[\sqrt{2\delta p_{\theta_{\text{old}}}(x)\left(1-p_{\theta_{\text{old}}}(x)\right)}\right]\right)\right)^2\right] \\
=&\mathbb{E}\left[\left(-\eta g^\top\hat{g}+\Delta_{\text{theo}}\right)^2\right] \\
=&\eta^2\mathbb{E}[(g^\top\hat{g})^2]-2\eta\Delta_{\text{theo}}\mathbb{E}[g^\top\hat{g}]+\Delta_{\text{theo}}^2
\end{aligned}
\tag{59}
$$

Because $\hat{\mathcal{L}}(\theta)$ is an unbiased estimator of $\mathcal{L}(\theta)$, we have

$$
\mathbb{E}\left[\nabla_\theta\hat{\mathcal{L}}(\theta)\right]=\nabla_\theta\mathbb{E}\left[\hat{\mathcal{L}}(\theta)\right]=\nabla_\theta\mathcal{L}(\theta)
\tag{60}
$$

$$
\mathbb{E}[\hat{g}]=\mathbb{E}\left[\nabla_\theta\hat{\mathcal{L}}(\theta)\big|_{\theta=\theta_{\text{old}}}\right]=\nabla_\theta\mathbb{E}\left[\hat{\mathcal{L}}(\theta)\big|_{\theta=\theta_{\text{old}}}\right]=\nabla_\theta\mathcal{L}(\theta)\big|_{\theta=\theta_{\text{old}}}=g
\tag{61}
$$

Therefore, we can obtain

$$
\mathbb{E}\left[g^\top\hat{g}\right]=g^\top\mathbb{E}\left[\hat{g}\right]=g^\top g
\tag{62}
$$

Now, regarding the first item:

$$
\mathbb{E}[(g^\top\hat{g})^2]=\mathbb{E}[g^\top\hat{g}\hat{g}^\top g]=g^\top\mathbb{E}[\hat{g}\hat{g}^\top]g
\tag{63}
$$

$$\mathbb{E}[\hat{g}\hat{g}^\top] = \mathbb{V}(\hat{g}) + \mathbb{E}[\hat{g}]\mathbb{E}[\hat{g}]^\top = \mathbb{V}(\hat{g}) + gg^\top \tag{64}$$

Hence, the original problem is equivalent to the following formulation:

$$\mathbb{E}\left[\left(\left(\mathcal{L}(\hat{\theta}) - \mathcal{L}(\theta_{\text{old}}) - \left(-\mathbb{E}_{x\sim\rho}\left[\sqrt{2\delta p_{\theta_{\text{old}}}(x)\left(1 - p_{\theta_{\text{old}}}(x)\right)}\right]\right)\right)^2\right]$$
$$=\eta^2\left(g^\top\mathbb{V}(\hat{g})g + \left(g^\top g\right)^2\right) - 2\eta\Delta_{\text{theo}}\left(g^\top g\right) + \Delta_{\text{theo}}^2 \tag{65}$$

Since we aim to minimize the gap from the theoretical update lower bound by reallocating the rollout quantities per question under a total sampling budget of $N$, the simplification of the objective function reveals that this problem only affects the first term $g^\top\mathbb{V}(\hat{g})g$. Thus, the original optimization problem is equivalent to the following:

$$\min g^\top\mathbb{V}(\hat{g})g, \quad \text{s.t. } \sum_{i=1}^m n_i = N. \tag{66}$$

The theoretical gradient direction $g$ is typically unknown, and we seek to control the uncertainty of the estimator in all possible directions. Therefore, we instead minimize the total variance $\text{Tr}(\mathbb{V}(\hat{g}))$, which corresponds to uniformly reducing the variance in all directions. This approach is a widely adopted technique for variance estimation(Bottou et al., 2018; Papini et al., 2018; Wang et al., 2013). In other words, we consider the following optimization problem:

$$\min \text{Tr}(\mathbb{V}(\hat{g})), \quad \text{s.t. } \sum_{i=1}^m n_i = N \tag{67}$$

Since each $y_j$ is independently draws from $\pi_{\theta_{\text{old}}}$, we can conclude that:

$$\mathbb{V}(\hat{g}) = \mathbb{V}\left(-\frac{1}{m}\sum_{i=1}^m\frac{1}{n_i}\sum_{y_j\in\mathcal{D}_i}\frac{\nabla_\theta\pi_\theta(y_j|x_i)\big|_{\theta=\theta_{\text{old}}}}{\pi_{\theta_{\text{old}}}(y_j|x_i)}A_{\theta_{\text{old}}}(x_i,y_j)\right)$$
$$=\frac{1}{m^2}\sum_{i=1}^m\mathbb{V}_{y_j\sim\pi_{\theta_{\text{old}}}}\left(\frac{1}{n_i}\sum_{y_j\in\mathcal{D}_i}\frac{\nabla_\theta\pi_\theta(y_j|x_i)\big|_{\theta=\theta_{\text{old}}}}{\pi_{\theta_{\text{old}}}(y_j|x_i)}A_{\theta_{\text{old}}}(x_i,y_j)\right) \tag{68}$$

Let $h(x,y;\theta) = \frac{\nabla_\theta\pi_\theta(y_j|x_i)}{\pi_{\theta_{\text{old}}}(y_j|x_i)}A_{\theta_{\text{old}}}(x_i,y_j)$, we have

$$\mathbb{V}(\hat{g}) =\frac{1}{m^2}\sum_{i=1}^m\mathbb{V}_{y_j\sim\pi_{\theta_{\text{old}}}}\left(\frac{1}{n_i}\sum_{j=1}^{n_i}h(y_j,x_i;\theta_{\text{old}})\right)$$
$$=\frac{1}{m^2}\sum_{i=1}^m\frac{1}{n_i^2}\cdot n_i\left(\mathbb{V}_{y\sim\pi_{\theta_{\text{old}}}}\left(h\left(y,x_i;\theta_{\text{old}}\right)\right)\right) \tag{69}$$
$$=\frac{1}{m^2}\sum_{i=1}^m\frac{\mathbb{V}_{y\sim\pi_{\theta_{\text{old}}}}\left(h\left(y,x_i;\theta_{\text{old}}\right)\right)}{n_i}$$

Therefore, for the total variance $\text{Tr}(\mathbb{V}(\hat{g}))$:

$$\text{Tr}(\mathbb{V}(\hat{g})) = \frac{1}{m^2}\sum_{i=1}^m\frac{\text{Tr}\left(\mathbb{V}_{y\sim\pi_{\theta_{\text{old}}}}\left(h\left(y,x_i;\theta_{\text{old}}\right)\right)\right)}{n_i}. \tag{70}$$

Then we dive into calculating the value of $\mathrm{Tr}\left(\mathbb{V}(h(y, x_i; \theta_{\mathrm{old}}))\right)$:

$$
\begin{aligned}
&\mathrm{Tr}\left(\mathbb{V}_{y \sim \pi_{\theta_{\mathrm{old}}}}(h(y, x_i; \theta_{\mathrm{old}}))\right) \\
=&\mathbb{E}_{y \sim \pi_{\theta_{\mathrm{old}}}}\left[\frac{\mathrm{Tr}\left(\nabla_\theta \pi_\theta(y|x_i) \nabla_\theta \pi_\theta(y|x_i)^\top \big|_{\theta=\theta_{\mathrm{old}}}\right)}{(\pi_{\theta_{\mathrm{old}}}(y|x_i))^2} A_{\theta_{\mathrm{old}}}^2\right] \\
&- \mathrm{Tr}\left(\mathbb{E}_{y \sim \pi_{\theta_{\mathrm{old}}}}\left[\left(\frac{\nabla_\theta \pi_\theta(y|x_i)\big|_{\theta=\theta_{\mathrm{old}}}}{\pi_{\theta_{\mathrm{old}}}(y|x_i)} A_{\theta_{\mathrm{old}}}\right)\right] \mathbb{E}_{y \sim \pi_{\theta_{\mathrm{old}}}}\left[\left(\frac{\nabla_\theta \pi_\theta(y|x_i)\big|_{\theta=\theta_{\mathrm{old}}}}{\pi_{\theta_{\mathrm{old}}}(y|x_i)} A_{\theta_{\mathrm{old}}}\right)\right]^\top\right) \\
=&\mathbb{E}_{y \sim \pi_{\theta_{\mathrm{old}}}}\left[\mathrm{Tr}\left(\nabla_\theta \log \pi_\theta(y|x_i) \nabla_\theta \log \pi_\theta(y|x_i)^\top \big|_{\theta=\theta_{\mathrm{old}}}\right) A_{\theta_{\mathrm{old}}}^2\right] \\
&- \mathrm{Tr}\left(\mathbb{E}_{y \sim \pi_{\theta_{\mathrm{old}}}}[\nabla_\theta \log \pi_\theta(y|x_i)\big|_{\theta=\theta_{\mathrm{old}}} A_{\theta_{\mathrm{old}}}] \mathbb{E}_{y \sim \pi_{\theta_{\mathrm{old}}}}[\nabla_\theta \log \pi_\theta(y|x_i)\big|_{\theta=\theta_{\mathrm{old}}} A_{\theta_{\mathrm{old}}}]^\top\right) \\
=&\mathbb{E}_{y \sim \pi_{\theta_{\mathrm{old}}}}\left[\mathrm{Tr}\left(\nabla_\theta \log \pi_\theta(y|x_i)^\top \big|_{\theta=\theta_{\mathrm{old}}} \nabla_\theta \log \pi_\theta(y|x_i)\big|_{\theta=\theta_{\mathrm{old}}}\right) A_{\theta_{\mathrm{old}}}^2\right] \\
&- \mathrm{Tr}\left(\mathbb{E}_{y \sim \pi_{\theta_{\mathrm{old}}}}[\nabla_\theta \log \pi_\theta(y|x_i)\big|_{\theta=\theta_{\mathrm{old}}} A_{\theta_{\mathrm{old}}}]^\top \mathbb{E}_{y \sim \pi_{\theta_{\mathrm{old}}}}[\nabla_\theta \log \pi_\theta(y|x_i)\big|_{\theta=\theta_{\mathrm{old}}} A_{\theta_{\mathrm{old}}}]\right) \\
=& \mathbb{E}_{y \sim \pi_{\theta_{\mathrm{old}}}}\left[\|\nabla_\theta \log \pi_\theta(y|x_i)\big|_{\theta=\theta_{\mathrm{old}}}\|^2 A_{\theta_{\mathrm{old}}}^2\right] - \left\|\mathbb{E}_{y \sim \pi_{\theta_{\mathrm{old}}}}[\nabla_\theta \log \pi_\theta(y|x_i) A_{\theta_{\mathrm{old}}}]\right\|^2
\end{aligned}
\tag{71}
$$

Consider the advantage function defined as:

$$
A_{\theta_{\mathrm{old}}}(x, y) = r(x, y) - \mathbb{E}_{y \sim \pi_{\theta_{\mathrm{old}}}}(r(x, y)).
\tag{72}
$$

We classify the rollouts into two categories based on whether the final answer is correct or not:

$$
\begin{aligned}
&\mathrm{Tr}\left(\mathbb{V}_{y \sim \pi_{\theta_{\mathrm{old}}}}(h(y, x_i; \theta_{\mathrm{old}}))\right) \\
=&P(r=1|x_i) \mathbb{E}_{y \sim \pi_{\theta_{\mathrm{old}}}, r=1}\left[\|\nabla_\theta \log \pi_\theta(y|x_i)\big|_{\theta=\theta_{\mathrm{old}}}\|^2 (1 - p_{\theta_{\mathrm{old}}}(x_i))^2\right] \\
&+ P(r=0|x_i) \mathbb{E}_{y \sim \pi_{\theta_{\mathrm{old}}}, r=0}\left[\|\nabla_\theta \log \pi_\theta(y|x_i)\big|_{\theta=\theta_{\mathrm{old}}}\|^2 (p_{\theta_{\mathrm{old}}}(x_i))^2\right] \\
&- \left\|P(r=1|x_i) \mathbb{E}_{y \sim \pi_{\theta_{\mathrm{old}}}, r=1}[\nabla_\theta \log \pi_\theta(y|x_i)(1 - p_{\theta_{\mathrm{old}}}(x_i)))]\right. \\
&\left.+ P(r=0|x_i) \mathbb{E}_{y \sim \pi_{\theta_{\mathrm{old}}}, r=0}[\nabla_\theta \log \pi_\theta(y|x_i)(-p_{\theta_{\mathrm{old}}}(x_i))]\right\|^2 \\
=&p_{\theta_{\mathrm{old}}}(x_i)(1 - p_{\theta_{\mathrm{old}}}(x_i))^2 \mathbb{E}_{y \sim \pi_{\theta_{\mathrm{old}}}, r=1}\left[\|\nabla_\theta \log \pi_\theta(y|x_i)\big|_{\theta=\theta_{\mathrm{old}}}\|^2\right] \\
&+ (p_{\theta_{\mathrm{old}}}(x_i))^2(1 - p_{\theta_{\mathrm{old}}}(x_i)) \mathbb{E}_{y \sim \pi_{\theta_{\mathrm{old}}}, r=0}\left[\|\nabla_\theta \log \pi_\theta(y|x_i)\big|_{\theta=\theta_{\mathrm{old}}}\|^2\right] \\
&- p_{\theta_{\mathrm{old}}}(x_i)^2(1 - p_{\theta_{\mathrm{old}}}(x_i))^2 \left\|\mathbb{E}_{y \sim \pi_{\theta_{\mathrm{old}}}, r=1}[\nabla_\theta \log \pi_\theta(y|x_i)] - \mathbb{E}_{y \sim \pi_{\theta_{\mathrm{old}}}, r=0}[\nabla_\theta \log \pi_\theta(y|x_i)]\right\|^2 \\
=& \sigma_i^2
\end{aligned}
\tag{73}
$$

Therefore, we need to solve the following problem:

$$
\min \frac{1}{m^2} \sum_{i=1}^m \frac{\sigma_i^2}{n_i}, \quad \text{s.t.} \sum_{i=1}^m n_i = N.
\tag{74}
$$

We also employ the Lagrange multiplier method to solve this problem:

$$
\min \mathcal{J} = \min \frac{1}{m^2} \sum_{i=1}^m \frac{\sigma_i^2}{n_i} + \mu(\sum_{i=1}^m n_i - N),
\tag{75}
$$

where $\mu$ is the Lagrange multiplier.

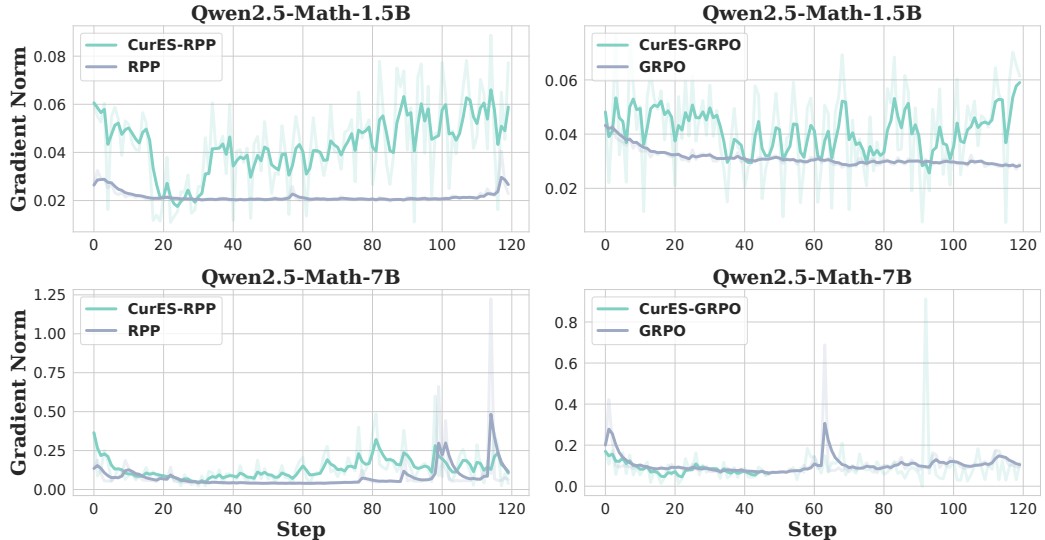

Figure 7: Comparison of Average Gradient Norms. This figure compares the average gradient norms among CurES-GRPO, CurES-RPP, GRPO, and RPP. The CurES variants consistently exhibit higher gradient norms in three out of the four algorithm-and-model-scale combinations, suggesting that the CurES effectively selects more informative prompts, thereby accelerating the training process.

By differentiating both sides with respect to $n_i$ and setting the derivative to 0, we obtain:

$$\frac{\partial \mathcal{J}}{\partial n_i} = -\frac{\sigma_i^2}{m^2 n_i^2} + \mu = 0 \implies n_i^2 = \frac{\sigma_i^2}{m^2 \mu}. \tag{76}$$

That is

$$n_i = \frac{\sigma_i}{m\sqrt{\mu}}. \tag{77}$$

According to the constraint:

$$\sum_{i=1}^{m} n_i = \sum_{i=1}^{m} \frac{\sigma_i}{m\sqrt{\mu}} = N \implies \sqrt{\mu} = \frac{\sum_{i=1}^{m} \sigma_i}{Nm}. \tag{78}$$

We get

$$n_i = \frac{\sigma_i}{\sum_j \sigma_j} N, \quad \sigma_i = \sqrt{\mathrm{Tr}\left(\mathbb{V}_{y \sim \pi_{\theta_{\mathrm{old}}}}\left(h\left(y, x_i; \theta_{\mathrm{old}}\right)\right)\right)}. \tag{79}$$

Thus, we derive the rollout quantity allocation strategy for different prompts.

## B   EXTENDED EXPERIMENTAL RESULTS

We further analyze the evolution of average gradient norms across different model scales and optimization algorithms (Figure 7). Overall, the CurES variants consistently exhibit stronger gradient signals compared to their corresponding baselines. On Qwen2.5-Math-1.5B, both CurES-RPP and CurES-GRPO maintain substantially higher gradient norms throughout training. On Qwen2.5-Math-7B, CurES-RPP continues to yield larger gradients, while CurES-GRPO performs comparably to GRPO. Higher gradient norms indicate that the model receives more informative learning signals, suggesting that CurES effectively prioritizes prompts that accelerate parameter updates. Notably, for the larger 7B model, the optimizer tends to dampen gradient magnitudes more significantly, which partially reduces the advantage of CurES; nevertheless, the overall trend demonstrates its robustness and consistent benefit across scales.

Figure 8 illustrates the distribution of rollout allocations across prompts with different accuracy levels over successive training iterations. We observe that CurES adaptively concentrates rollouts on

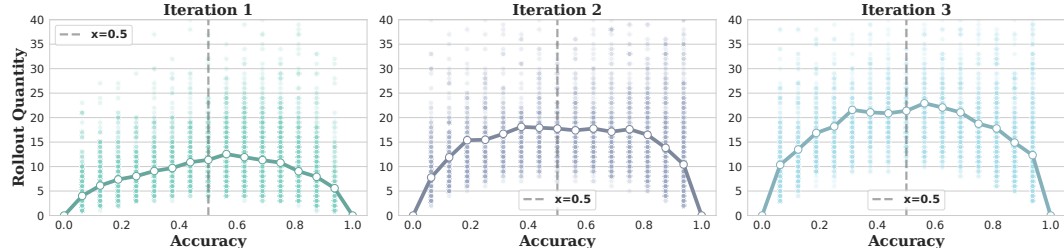

Figure 8: Distribution of rollout quantities with respect to accuracy in CurES base on Qwen2.5-Math-7B at different training iterations. CurES concentrates more rollouts on moderately difficult prompts.

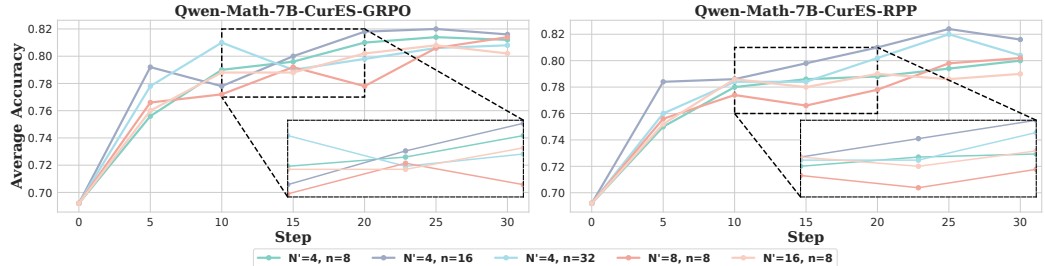

Figure 9: Performance convergence of Qwen2.5-Math-CurES-7B on MATH500 with different sampling configurations.

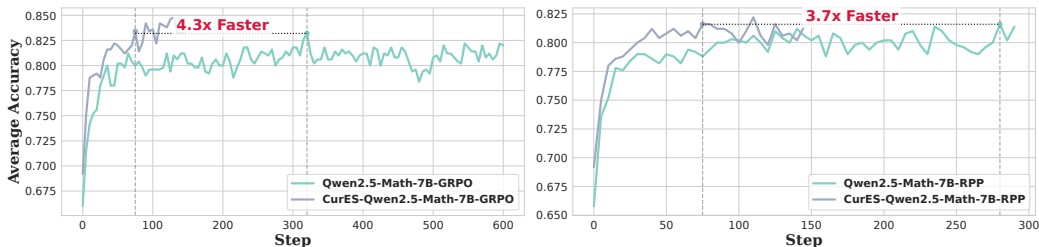

Figure 10: Efficiency comparison of CurES against baselines on MATH500 with Qwen2.5-Math-7B. Gray dashed lines indicate the steps required for CurES and the baseline to reach the highest average accuracy within the total training duration.

moderately difficult prompts, rather than uniformly sampling across the entire spectrum. This behavior aligns with the intuition that prompts with intermediate difficulty provide the most informative learning signal—being neither trivially solved nor consistently incorrect. As training progresses, the distribution becomes increasingly peaked around this region, indicating that CurES dynamically refines its sampling strategy to focus computational effort on prompts that are most beneficial for improving policy performance.

We further present the performance convergence of CurES with different sampling configurations on Qwen2.5-Math-7B, as shown in Figure 9. We observe that CurES achieves stable improvements across all settings, with only minor differences in convergence speed and final accuracy among varying rollout counts and prompt subsets. Notably, configurations with a moderate number of rollouts (e.g., $N' = 4, n = 16$) strike a favorable balance, reaching higher accuracy with fewer steps compared to more extreme settings such as very large or very small rollout numbers. This demonstrates that CurES is robust to sampling configurations and can effectively leverage diverse rollout budgets without significant degradation in performance.

We also compare the training efficiency of CurES against the baselines on MATH500 for 7B models. As shown in Figure 10, CurES reaches the highest accuracy achieved by the baselines 4.3× faster under GRPO and 3.7× faster under RPP. Importantly, CurES not only converges more rapidly but

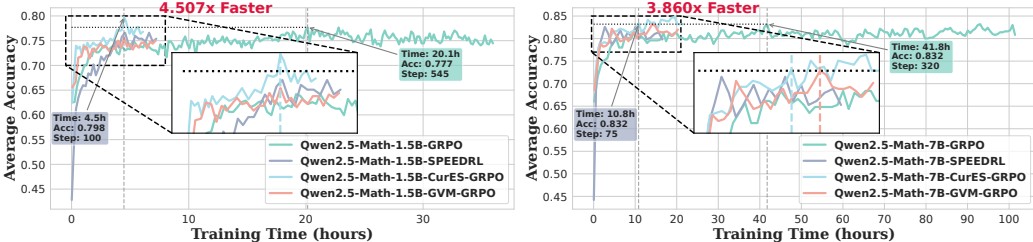

Figure 11: Validation performance with training time of CurES against baselines on MATH500. The gray dashed lines mark the training time CurES and each baseline required to reach the baselines' highest average accuracy within the total training duration.

Table 2: Ablation study on isolating the contributions of prompt sampling and rollout allocation based on the Qwen2.5-Math-1.5B model.

| Method | Pass@1 | | | | | Average@16 | | | Avg. |
|---|---|---|---|---|---|---|---|---|---|
| | MATH500 | GSM8K | GAO23 | MINERVA | OLYM | AIME24 | AIME25 | AMC23 | |
| Qwen2.5-Math-1.5B | 40.20 | 43.90 | 25.19 | 11.40 | 21.04 | 1.67 | 1.67 | 14.84 | 20.00 |
| +CurES-GRPO | 77.20 | 85.97 | 51.43 | 31.62 | 37.33 | 13.33 | 10.42 | 52.19 | 44.94 |
| +CurES-GRPO w/o Rollout Allocation | 76.20 | 84.38 | 49.61 | 29.04 | 36.30 | 13.33 | 9.38 | 51.41 | 43.71 |
| +CurES-GRPO w/o Prompt Sampling | 77.00 | 84.00 | 50.25 | 29.04 | 36.59 | 13.33 | 11.46 | 50.94 | 44.08 |

Table 3: Additional baseline comparison with DAPO (Yu et al., 2025) and MoPPS (Qu et al., 2025a) based on the Qwen2.5-Math-1.5B model.

| Method | Pass@1 | | | | | Average@16 | | | Avg. |
|---|---|---|---|---|---|---|---|---|---|
| | MATH500 | GSM8K | GAO23 | MINERVA | OLYM | AIME24 | AIME25 | AMC23 | |
| Qwen2.5-Math-1.5B | 40.20 | 43.90 | 25.19 | 11.40 | 21.04 | 1.67 | 1.67 | 14.84 | 20.00 |
| +DAPO | 71.40 | 86.28 | 49.61 | 27.94 | 35.26 | 12.50 | 7.71 | 51.72 | 42.80 |
| +MoPPS | 72.64 | 84.23 | 50.25 | 31.62 | 37.33 | 12.92 | 10.83 | 52.03 | 43.98 |
| +CurES-GRPO | 77.20 | 85.97 | 51.43 | 31.62 | 37.33 | 13.33 | 10.42 | 52.19 | 44.94 |

also exceeding the baselines throughout training. These findings confirm that the adaptive prompt selection mechanism of CurES significantly improves sample efficiency, enabling faster convergence and better final model quality.

Moreover, the results in Figure 11 demonstrate that CurES significantly accelerates convergence without compromising model performance. As indicated by the gray dashed lines—which mark the time required to reach the baselines' peak accuracy—CurES achieves substantial speedups. Specifically, for the 1.5B model, CurES attains the target accuracy approximately 4.51x faster than the GRPO baseline (reducing the required time from 20.1 hours to just 4.5 hours). Similarly, for the 7B model, CurES delivers a 3.86x speedup, reaching the performance threshold in 10.8 hours compared to the baseline's 41.8 hours.

Table 2 presents the ablation results on the Qwen2.5-Math-1.5B model, where we independently evaluate the contributions of the two key components in CurES: prompt sampling strategy and roll-out allocation mechanism. The empirical results demonstrate that both components are essential to the overall performance gains. In particular, removing either the difficulty-aware prompt sampling strategy or the adaptive rollout allocation mechanism leads to a notable decrease in overall accuracy, thereby validating their complementary contributions to the effectiveness of CurES.

Table 3 demonstrates the performance comparison of our method with DAPO (Yu et al., 2025) and MoPPS (Wang et al., 2025) on the Qwen2.5-Math-1.5B model. While both DAPO and MoPPS demonstrate notable improvements over the base model, CurES-GRPO achieves superior overall performance, indicating that our gradient-based framework provides a more effective mechanism for accelerating training.

Table 4 reports the experimental results across different model architectures, where we additionally evaluate our method on Llama3.2-3B-Instruct and Phi-4-mini-Instruct. Our findings demonstrate that CurES-GRPO consistently outperforms vanilla GRPO in both model families. This consistent

Table 4: Performance comparison of different methods on Llama3.2-3B-Instruct and Phi-4-mini-Instruct.

| Method | Pass@1 | | | | | Average@16 | | | Avg. |
|---|---|---|---|---|---|---|---|---|---|
| | MATH500 | GSM8K | GAO23 | MINERVA | OLYM | AIME24 | AIME25 | AMC23 | |
| Llama3.2-3B-Instruct | 29.38 | 78.17 | 38.96 | 18.22 | 14.52 | 5.62 | 0.00 | 20.31 | 25.65 |
| +GRPO | 52.52 | 80.44 | 40.26 | 19.49 | 18.81 | 9.19 | 0.42 | 29.22 | 31.29 |
| +CurES-GRPO | 60.40 | 83.09 | 42.08 | 21.69 | 21.96 | 12.29 | 0.21 | 32.66 | 34.30 |
| Phi-4-mini-Instruct | 65.39 | 85.06 | 40.52 | 18.75 | 19.70 | 2.08 | 0.21 | 21.09 | 31.60 |
| +GRPO | 67.81 | 88.32 | 41.04 | 36.40 | 30.81 | 7.08 | 2.29 | 38.75 | 39.06 |
| +CurES-GRPO | 76.80 | 87.87 | 42.86 | 35.74 | 31.11 | 7.71 | 3.12 | 35.78 | 40.12 |

enhancement confirms the robustness and generalizability of the CurES framework across diverse model architectures.

---

**Case Study 1**

**Problem** Janet goes to the mall and spends $3.50 on ice cream, $7.50 each for movie tickets for herself and her younger sister, and $8.50 on a bracelet. If her parents gave her $40, how many dollars does she have left?

**Answer:** 13

**GVM Output** First, let's calculate the total amount of money Janet spent. She spent $3.50 on ice cream, $7.50 on a movie ticket for herself, $7.50 on a movie ticket for her younger sister, and $8.50 on a bracelet. So the total amount spent is: [3.50 + 7.50 + 7.50 + 8.50] Next, we need to find out how much money she has left after spending this amount. Her parents gave her $40, so the remaining amount is: [40 - (3.50 + 7.50 + 7.50 + 8.50)] Let's calculate this step by step using Python.

```python
# Amount spent on ice cream
ice-cream = 3.50
# Amount spent on movie tickets for herself and her younger sister
movie-tickets = 7.50 + 7.50
# Amount spent on bracelet
bracelet = 8.50
# Total amount spent
total-spent = ice-cream + movie-tickets + bracelet
# Amount given by parents
amount-given = 40.00
# Amount left
amount-left = amount-given - total-spent
print(amount-left)
```

```output
14.0
```

After calculating, we find that Janet has  14  dollars left.

**CurES Output** Let's break down the problem step by step and then use Python to verify our calculations.

1. First, we need to calculate the total amount Janet spent:

- $3.50 on ice cream
- $7.50 each for movie tickets for herself and her younger sister (so $7.50 + $7.50 = $15.00)
- $8.50 on a bracelet

So, the total amount spent is $3.50 + $15.00 + $8.50 = $27.00.

2. Janet's parents gave her $40. To find out how much money she has left, we subtract the total amount spent from the $40 her parents gave her: [40 − 27 = 13]

The calculations confirm that Janet has  13  dollars left.

---

**Case Study 2**

**Problem** The Sun's effective temperature, $T_e$, is 5800 Kelvin, and its radius is $7 \times 10^{10}$ cm $\left(7 \times 10^8 m\right)$. Compute the luminosity (power output) of the Sun in erg/s. Treat the Sun as a blackbody radiator, and give your answer to one significant figure.

**Answer:** $4 \times 10^{33}$

**GVM Output** The luminosity $L$ of a blackbody can be calculated using the Stefan-Boltzmann law:

$$L = 4\pi R^2 \sigma T_e^4$$

where:

- $R$ is the radius,
- $T_e$ is the effective temperature,
- $\sigma$ is the Stefan-Boltzmann constant, approximately $5.67 \times 10^{-8} \mathrm{W} \cdot \mathrm{m}^{-2} \cdot \mathrm{K}^{-4}$.

First, convert the radius from cm to meters:

$$R = 7 \times 10^{10} \mathrm{cm} = 7 \times 10^8 \mathrm{m}$$

The effective temperature is given as:

$$T_e = 5800 \mathrm{K}$$

Now plug the values into the formula:

$$L = 4\pi (7 \times 10^8)^2 (5.67 \times 10^{-8})(5800)^4$$

Calculate $R^2$:

$$R^2 = (7 \times 10^8)^2 = 49 \times 10^{16} = 4.9 \times 10^{17} \mathrm{m}^2$$

Calculate $T_e^4$:

$$T_e^4 = (5800)^4 = 1.1316496 \times 10^{14} \mathrm{K}^4$$

Substitute the intermediate results back into the luminosity equation:

$$L = 4\pi (4.9 \times 10^{17})(5.67 \times 10^{-8})(1.1316496 \times 10^{14})$$

Combine the constants and the powers of 10:

$$L = 4\pi (4.9 \times 5.67 \times 1.1316496 \times 10^{17-8+14}) = 4\pi (4.9 \times 5.67 \times 1.1316496 \times 10^{23})$$

Calculate the product of the constants:

$$4.9 \times 5.67 \times 1.1316496 \approx 31.03$$

So,

$$L \approx 4\pi (31.03 \times 10^{23}) = 4\pi (3.103 \times 10^{24})$$

Now compute the numerical value:

$$L \approx 4 \times 3.14159 \times 3.103 \times 10^{24} \approx 39.4784 \times 3.103 \times 10^{24} \approx 122.5 \times 10^{24}$$

$$L \approx 1.225 \times 10^{26} \mathrm{W}$$

Rounded to one significant figure:

$$L \approx 1 \times 10^{26} \mathrm{W}$$

Since $1 \mathrm{W} = 1 \mathrm{J/s} = 10^7 \mathrm{erg/s}$:

$$L \approx 1 \times 10^{26} \times 10^7 \mathrm{erg/s} = 1 \times 10^{33} \mathrm{erg/s}$$

The luminosity in erg/s is:

$$\boxed{1 \times 10^{33}}$$

**CurES Output** The luminosity $L$ of a blackbody is given by the Stefan-Boltzmann law:

$$L = 4\pi R^2 \sigma T_e^4$$

where:

- $R$ is the radius of the Sun,
- $T_e$ is the effective temperature of the Sun,
- $\sigma$ is the Stefan-Boltzmann constant, $\sigma \approx 5.67 \times 10^{-8} \mathrm{W} \cdot \mathrm{m}^{-2} \cdot \mathrm{K}^{-4}$.

Convert the solar radius from centimeters to meters:

$$R = 7 \times 10^{10} \mathrm{cm} = 7 \times 10^8 \mathrm{m}$$

The effective temperature is:

$$T_e = 5800 \mathrm{K}$$

$$L = 4\pi (7 \times 10^8)^2 \times (5.67 \times 10^{-8}) \times (5800)^4$$

Calculate $R^2$:

$$R^2 = (7 \times 10^8)^2 = 49 \times 10^{16} = 4.9 \times 10^{17} \mathrm{m}^2$$

Calculate $T_e^4$:

$$T_e^4 = (5800)^4 = 1.1316496 \times 10^{14} \mathrm{K}^4$$

Substitute the intermediate results:

$$L = 4\pi \times (4.9 \times 10^{17}) \times (5.67 \times 10^{-8}) \times (1.1316496 \times 10^{14})$$

Combine the constants and the powers of 10:

$$L = 4\pi \times (4.9 \times 5.67 \times 1.1316496) \times 10^{17-8+14} = 4\pi \times (31.37336) \times 10^{23} \mathrm{W}$$

$$L \approx 4 \times 3.14159 \times 31.37336 \times 10^{23} \approx 394.444 \times 10^{23} \mathrm{W} \approx 3.94444 \times 10^{25} \mathrm{W}$$

Rounded to one significant figure:

$$L \approx 4 \times 10^{26} \mathrm{W}$$

Since $1\mathrm{W} = 1\mathrm{J/s} = 10^7 \mathrm{erg/s}$:

$$L = 4 \times 10^{26} \times 10^7 \mathrm{erg/s} = 4 \times 10^{33} \mathrm{erg/s}$$

The luminosity of the Sun is:

$$\boxed{4 \times 10^{33}} \mathrm{erg/s}$$

## C    USE OF LLMS

We use LLMs only to refine the language and grammar in our paper. We do not use them for generating research ideas or for finding related work. We provide our complete original text to OpenAI's GPT-4o with instructions to make it more professional, coherent, and native-sounding for a research paper. We then carefully review all suggestions to guarantee that no factual content is altered and that all changes remain true to our original writing.

