# OpenReview forum: "CurES: From Gradient Analysis to Efficient Curriculum Learning for Reasoning LLMs"
_ICLR.cc/2026/Conference — ICLR 2026 Poster_

### Official Review · Reviewer_9BeW · 2025-10-30

**Soundness:** 3
**Presentation:** 3
**Contribution:** 3
**Rating:** 6
**Confidence:** 4

**Summary:**

This paper studies curriculum learning for RL finetuning of reasoning LLMs under a verifiable-reward setting. The authors analyze how different prompts contribute to learning progress by examining the relationship between success probability and optimization potential. They find that prompts with intermediate accuracy are the most informative and propose CurES, a principled task sampling and rollout allocation framework. CurES estimates each prompt’s accuracy using a Beta posterior, samples prompts according to a concave transformation of their estimated uncertainty, and allocates rollout counts to minimize gradient variance. The method is implemented as a multi-iteration procedure that periodically refreshes estimates to adapt to distributional shifts. Experiments using Qwen2.5-Math models show that CurES achieves faster convergence and higher final accuracy than GRPO, GVM, Speed-RL, and RPP.

**Strengths:**

- The paper provides a principled derivation linking sampling and rollout allocation to expected loss change and gradient variance, and successfully turns the analysis into a practical algorithm.
- The paper is well-structured.

**Weaknesses:**

- Narrow experimental scope. The training dataset is limited to Numina-Math, which limits evidence of general applicability to other task types or reward settings.
- Limited ablation studies. The paper does not include results for partial variants that use only task sampling or only rollout allocation, making it unclear how much each component contributes individually.
- Incomplete baseline comparison. Some relevant online data filtering or adaptive sampling methods, such as DAPO[1] and MoPPS[2], are mentioned but not compared experimentally. Moreover, the proposed Prompt Difficulty Assessment appears conceptually similar to MoPPS, and the paper should clearly articulate the differences. Additionally, utilizing optimization histories to guide efficient sampling shares similarities with Bayesian-based active sampling methods[3,4], which warrants further discussion in related work.
- Lack of discussion on prior theoretical overlap. The theoretical finding that tasks with intermediate success probability are the most informative has been observed or hypothesized in several prior works[5,6,7,8]. The paper should acknowledge these connections and clarify how its derivation provides new insight beyond existing results.
- Algorithmic clarity. Algorithm 1 is somewhat confusing, especially regarding the role and purpose of the pre-rollout stage, which requires additional explanation.
- Limited model diversity. All experiments are conducted on Qwen2.5-Math models, leaving uncertainty about whether the method generalizes to other model families such as LLaMA or Mistral.

[1] Dapo: An open-source llm reinforcement learning system at scale.\
[2] Can prompt difficulty be online predicted for accelerating rl finetuning of reasoning models?\
[3] Model Predictive Task Sampling for Efficient and Robust Adaptation\
[4] Fast and Robust: Task Sampling with Posterior and Diversity Synergies for Adaptive Decision-Makers in Randomized Environments\
[5] Self-Evolving Curriculum for LLM Reasoning\
[6] Online Difficulty Filtering for Reasoning Oriented Reinforcement Learning\
[7] Not All Rollouts are Useful: Down-Sampling Rollouts in LLM Reinforcement Learning\
[8] SPEED-RL: Faster Training of Reasoning Models via Online Curriculum Learning

**Questions:**

see weaknesses

---

> ### Author Response · Authors · 2025-11-25
> **Rebuttal to W1-W2**
>
> **Dear Reviewer 9BeW**,
>
> We extend our sincere gratitude to the reviewer for the constructive comments and suggestions. Below is our response to your concerns:
>
> >**[W1]: Narrow experimental scope. The training dataset is limited to Numina-Math, which limits evidence of general applicability to other task types or reward settings.**
>
> **W1-A1**: We sincerely appreciate the reviewer’s valuable suggestion regarding the experimental scope. Our design prioritizes the **controllability and reproducibility** of the training environment. The decision to evaluate on **Numina-Math** was made to strictly align with the experimental setup of the recent state-of-the-art method **GVM [1]**. This choice facilitates a **direct and fair comparison** between approaches under consistent reward structures and difficulty conditions.
>
> We fully concur with the reviewer on the importance of expanding the scope of experiments. However, given the **time constraints of the rebuttal period**, we have made every effort to conduct the supplementary experiments using our available resources. We plan to extend our method to additional task domains (such as coding and planning) in future work to further assess the generality of our approach.
>
>
> **References**
>
> [1] Optimizing Chain-of-Thought Reasoners via Gradient Variance Minimization in Rejection Sampling and RL, NeurIPS 2025.
>
>
>
> >**[W2]: Limited ablation studies.**
>
> **W2-A2**: To evaluate the **individual contributions** of each module, we conducted ablation studies using the **Qwen2.5-Math-1.5B** model. Specifically, we isolated the two core components of **CurES**: **prompt difficulty–based sampling** and **rollout allocation**. The experimental results are presented below:
>
> | Method  | MATH500 | GSM8K | GAO23 | MINERVA | OLYM  | AIME24 | AIME25 | AMC23 | Avg.  |
> | ---------------------------------- | ------- | ----- | ----- | ------- | ----- | ------ | ------ | ----- | ----- |
> | Qwen2.5-Math-1.5B                  | 40.20   | 43.90 | 25.19 | 11.40   | 21.04 | 1.67   | 1.67   | 14.84 | 20.00 |
> | +CurES-GRPO                        | 77.20   | 85.97 | 51.43 | 31.62   | 37.33 | 13.33  | 10.42  | 52.19 | 44.94 |
> | +CurES-GRPO w/o Rollout Allocation | 76.20   | 84.38 | 49.61 | 29.04   | 36.30 | 13.33  | 9.38   | 51.41 | 43.71 |
> | +CurES-GRPO w/o Prompt Sampling    | 77.00   | 84.00 | 50.25 | 29.04   | 36.59 | 13.33  | 11.46  | 50.94 | 44.08 |
>
> These empirical findings demonstrate that **both components are integral** to the achieved performance gains. Specifically, the exclusion of either **difficulty-based sampling** or **rollout allocation** results in a **measurable degradation** in overall accuracy, thereby substantiating their **complementary roles** within the CurES framework.

---

> ### Author Response · Authors · 2025-11-25
> **Rebuttal to W3**
>
> >**[W3]: Incomplete baseline comparison.**
>
> **W3-A3**: We appreciate this constructive suggestion. To **fully address** this concern, we conducted **additional experiments** with **DAPO [1]** and **MoPPS [2]** using the **Qwen2.5-Math-1.5B** model, and the results are detailed below:
>
> | Method            | MATH500 | GSM8K | GAO23 | MINERVA | OLYM  | AIME24 | AIME25 | AMC23 | Avg.  |
> | ----------------- | ------- | ----- | ----- | ------- | ----- | ------ | ------ | ----- | ----- |
> | Qwen2.5-Math-1.5B | 40.20   | 43.90 | 25.19 | 11.40   | 21.04 | 1.67   | 1.67   | 14.84 | 20.00 |
> | +DAPO             | 71.40   | 86.28 | 49.61 | 27.94   | 35.26 | 12.50  | 7.71   | 51.72 | 42.80 |
> | +MoPPS            | 72.64   | 84.23 | 50.25 | 31.62   | 37.33 | 12.92  | 10.83  | 52.03 | 43.98 |
> | +CurES-GRPO       | 77.20   | 85.97 | 51.43 | 31.62   | 37.33 | 13.33  | 10.42  | 52.19 | 44.94 |
>
> While **DAPO [1]** and **MoPPS [2]** yield notable improvements, **CurES-GRPO achieves superior overall performance**, demonstrating that our **gradient-based approach** offers a more effective mechanism for accelerating training.
>
> Regarding **MoPPS [2]**, we acknowledge it as a significant and elegant work and wish to clarify the **fundamental conceptual differences**:
>
> *   **MoPPS** focuses on online prediction of prompt difficulty to accelerate model training and reduce cost. Its core lies in the learning and updating process of a difficulty predictor throughout training.
> *   Our method, **CurES** is fundamentally motivated by **gradient-based optimization** for training efficiency. The core of our approach lies in the gradient-driven optimization of prompt sampling distribution and rollout allocation. The difficulty estimation module in Section 4.3 is not a standalone difficulty-prediction mechanism; rather, it serves as a **supporting component** to **instantiate the theoretically derived sampling distribution (Eq.14)** and **rollout assignment strategy (Eq.20)**.
> *   Under this formulation, our approach differs significantly in the **utilization of difficulty estimation**, the **mitigation strategy of distribution shift** in difficulty estimation caused by policy updates, and the **integration of difficulty estimation with gradient-based optimization**.
>
> Finally, we thank the reviewer for highlighting **MPTS [3]** and **PDTS [4]**. These works **leverage posteriors and optimization histories for task selection and active sampling**, which share conceptual connections with our work. **We have cited these works** and incorporated a more fine-grained discussion in the revised Related Work section.

---

> ### Author Response · Authors · 2025-11-25
> **Rebuttal to W4-W5**
>
> >**[W4]: Lack of discussion on prior theoretical overlap.**
>
> **W4-A4**: We sincerely appreciate this insightful comment. We acknowledge that prior works [5,6,7,8] have also similarly observed or hypothesized that samples of intermediate success probability are the most informative. These studies capture similar phenomena through empirical evidence or heuristic formulations. We will explicitly highlight this theoretical alignment and include these references in the revised manuscript.
>
> Concurrently, we would like to clarify the relationship between CurES and these works, highlighting the **distinct theoretical contributions** of our approach:
>
> *   **SEC [5]** categorizes the dataset and applies weighting based on estimated advantage $\hat{A}$ within each category, which **remains a relatively coarse-grained weighting approach.**
> *   **PODS [7]** downsamples rollouts by maximizing reward distribution variance, mitigating the compute and memory asymmetry between rollout generation and policy updates. However, **its theoretical justification does not explicitly derive a mathematical linkage between prompt difficulty and gradient efficiency, and thus remains an implicitly correlated formulation.**
> *   **Online Difficulty Filtering [6]** and **SPEED-RL [8]** respectively interpret the significance of moderate-difficulty samples from the perspectives of KL divergence and signal-to-noise ratio, yet both adopt a **generate-then-filter pipeline** for **prompt selection**.
>
> In contrast, our method CurES provides a more **systematic theoretical framework**. We are the **first to explicitly derive** the intrinsic relationship between **problem difficulty and gradient efficiency from a gradient-based perspective (Eq. 11)**, mathematically proving how difficulty influences the rate of gradient descent. Furthermore, we bridge the gap between theory and practical training efficiency through an **optimized rollout allocation formulation (Eq. 20)**. Thus, in CurES, the focus on intermediate difficulty is not a heuristic artifact or empirical intuition but emerges directly from a **provable theoretical foundation**, which we operationalize into a **practical curriculum strategy** for training.
>
> **References**
>
> [1] Dapo: An open-source llm reinforcement learning system at scale.
>
> [2] Can prompt difficulty be online predicted for accelerating rl finetuning of reasoning models?
>
> [3] Model Predictive Task Sampling for Efficient and Robust Adaptation
>
> [4] Fast and Robust: Task Sampling with Posterior and Diversity Synergies for Adaptive Decision-Makers in Randomized Environments
>
> [5] Self-Evolving Curriculum for LLM Reasoning
>
> [6] Online Difficulty Filtering for Reasoning Oriented Reinforcement Learning
>
> [7] Not All Rollouts are Useful: Down-Sampling Rollouts in LLM Reinforcement Learning
>
> [8] SPEED-RL: Faster Training of Reasoning Models via Online Curriculum Learning
>
> >**[W5]: Algorithmic clarity.**
>
> **W5-A5**: To further enhance clarity, we have integrated Algorithm 1 into the main text and added a **corresponding illustration** in the revised manuscript to better depict the process.
>
> Specifically, our proposed algorithm operates as a **two-stage procedure**:
>
> 1.  **Parameter Estimation Stage** (pre-rollout stage): We perform a limited number of rollouts (parameter estimation sample
> size $N'$, set to 4 in our experiments) to **estimate the initial prompt difficulty and gradient variance relative to the current model policy.**
> 2.  **Training Stage**: Leveraging these initial estimates, we proceed to the main training phase. In this stage, we first sample $m$ prompts according to Eq. (14) to form a prompt batch, and then collect a total of $N$ rollouts for this batch following our rollout allocation strategy specified in Eq. (20). This implies that, on average, each prompt receives $n$ rollouts, satisfying $N = m \times n$.

---

> ### Author Response · Authors · 2025-11-25
> **Rebuttal to W6**
>
> >**[W6]: Limited model diversity.**
>
> **W6-A6**: To alleviate the reviewer's concern, we additionally conduct experiments on the LLaMA and Phi model families, and the results are as follows:
>
> | Method                | MATH500 | GSM8K | GAO23 | MINERVA | OLYM  | AIME24 | AIME25 | AMC23 | Avg.  |
> | --------------------- | ------- | ----- | ----- | ------- | ----- | ------ | ------ | ----- | ----- |
> | LlamMA3.2-3B-Instruct | 29.38   | 78.17 | 38.96 | 18.22   | 14.52 | 5.62   | 0.00   | 20.31 | 25.65 |
> | +GRPO                 | 52.52   | 80.44 | 40.26 | 19.49   | 18.81 | 9.19   | 0.42   | 29.22 | 31.29 |
> | +CurES-GRPO           | 60.40   | 83.09 | 42.08 | 21.69   | 21.96 | 12.29  | 0.21   | 32.66 | 34.30 |
> | Phi-4-mini-Instruct   | 65.39   | 85.06 | 40.52 | 18.75   | 19.70 | 2.08   | 0.21   | 21.09 | 31.60 |
> | +GRPO                 | 67.81   | 88.32 | 41.04 | 36.40   | 30.81 | 7.08   | 2.29   | 38.75 | 39.06 |
> | +CurES-GRPO           | 76.80   | 87.87 | 42.86 | 35.74   | 31.11 | 7.71   | 3.12   | 35.78 | 40.12 |
>
> Across both the LLaMA and Phi model families, CurES-GRPO consistently outperforms vanilla GRPO, demonstrating the robustness and generality of our method across diverse model architectures.
>
> ----
> Finally, we sincerely thank the reviewer for the constructive and insightful comments. We will carefully take all of the feedback into account to further improve the clarity and quality of the manuscript.

---

### Official Review · Reviewer_Lzg2 · 2025-10-30

**Soundness:** 2
**Presentation:** 3
**Contribution:** 3
**Rating:** 6
**Confidence:** 3

**Summary:**

This paper discusses curriculum learning for training Large Language Models with Reinforcement Learning (RL) for reasoning tasks. From gradient analysis, the authors demonstrate the intrinsic relationship between optimization efficiency and prompt sampling distribution, as well as rollout quantities. Therefore, an curriculum learning algorithm called CurES is proposed to adjust the sampling probability and rollout quantities based on the prompt difficulty. Experiments show that CurES improves the efficiency and stability of GRPO across several math reasoning tasks.

**Strengths:**

1. The paper is clearly written and well-motivated.
2. The theoretical analysis is interesting and aligns with people's understanding on RLVR, Eq (11) shows that training with a prompt with higher variance is more helpful; while Equation (20) indicate that more rollout should be allocated to the prompts with higher variance of gradients.
3. The assumption of Beta-distribution in Section 4.3 address a good trade-off between theory and practice, leading to a simple yet practical algorithm.
4. From empirical results, CurES outperforms other baselines.
5. Comprehensive experiments demonstrate the effectiveness of proposed algorithm compared to baselines

**Weaknesses:**

## Section 4.1: Mismatched derivation from single prompt to dataset-level behavior.

From Eq (3) to Eq (11), the author analysis the impact of $|\mathcal{L}(x; \theta_{old}+d) - \mathcal{L}(x;\theta_{old})|$ to the variance of $r(x, y)$ based on Cramér–Rao inequality, which indicates that training on prompts with higher variance is more helpful. This is an interesting and meaningful observation. However, from Eq (11) to Eq (12), the derivation jumps from $|\mathcal{L}(x; \theta_{old}+d) - \mathcal{L}(x;\theta_{old})|$ to $|\mathcal{L}(\theta_{old}+d) - \mathcal{L}(\theta_{old})|$, which is actually **mismatched in the parameter update $d$**:
in the first expression $d$ denotes the parameter update w.r.t. a **single prompt** $x$, while in the second it implicitly refers to the **dataset-level** update. Ideally, the second one should be replaced by
$\bar{d} = \mathbb{E}_{x\sim\rho}[d]$. This implies that:

1) $d \neq \bar{d}$, ; therefore, it is not mathematically correct to generalize directly from a single prompt to the dataset average.

2) $\bar{d}$ also depends on your sampling distribution $\rho$, which is ignored.

Unfortunately, all the subsequent derivations are built upon this mismatch.

**Q1**: How can the authors justify this mismatch between $d$ and $\bar{d}$? What's its influence to the analysis afterwards?

**Q2**: Is it accurate to describe the variance term $\sqrt{2δp_{θ_{old}} (x) (1 − p_{θ_{old}} (x))}$ by prompt difficulty? It's more related to the variance. For example, both too difficult and too simple prompt could have zero variance.

## Section 4.2: Unexplained objective and beyond

In Section 4.2, the authors propose to optimize the objective Eq (15), which is $\mathbb{E}[(\mathcal{L}(\hat{\theta}) - \mathcal{L}(\theta_{old})  - \Delta_{theo})^2 ]$ (according to Eq (58)). This objective is the **Mean-Square Error (MSE)** between *actual improvement* $\mathcal{L}(\hat{\theta}) - \mathcal{L}(\theta_{old})$ and *theoretical improvement* $\Delta_{theo}$. However, the paper does not provide sufficient intuition for why minimizing this MSE corresponds to a principled curriculum-learning objective under a rollout budget constraint.

**Q3**: Why should Eq. (15) be viewed as a valid curriculum-learning objective? What is the intuition?

**Q4**: Since Eq. (15) builds on Section 4.1, it inherits the mismatched $d–\bar{d}$ issue. How does that affect the soundness of the subsequent results?

**Q5**: Moreover, the advantage estimation $A_{\theta_{old}}(x_i, y_i)$ also depends on the rollout number $n_i$ (and also the underlying policy gradient algorithm). This is ignored during the analysis.

Moving forward, the authors derive an optimal allocation of rollouts in Eq (20), i.e., $n_i = \frac{\sigma_i}{\sum_j \sigma_j} N$, where $\sigma_i$ is the variance of gradient of $x_i$ w.r.t $y$. Although the authors derive a symmetric computational form in Eq (21), it still seems computational intractable, especially for large langauge models. Concretely, the term $\mathbb{E}{y\sim\pi_{\theta_{\text{old}}}}\big[||\nabla_\theta \log\pi_\theta(y|x_i)||^2\big]$ requires computing the gradient norm w.r.t **each response $y_i$**, taking $N$ backward in total, while normal RL algorithm only needs to compute the average gradient once per batch.

**Q6**: What's the **time** and **memory** overhead of CurES, especially on estimating $\sigma_i$? Currently this part is missing in the Experiment Section.

## Other question

**Q7**: What’s the performance of baseline algorithms with different sample budget? Figure 5 only shows the performance of CurES.

**Questions:**

Please refer to the "Weakness"

---

> ### Author Response · Authors · 2025-11-25
> **Rebuttal to W1**
>
> **Dear Reviewer Lzg2**:
>
> We extend our sincere gratitude to the reviewer for the constructive comments and suggestions. Below is our response to your concerns:
>
> **[W1]: Mismatched derivation from single prompt to dataset-level behavior.**
>
> **The derivation from Eq (11) to Eq (12), identifying a mismatch in the parameter update $d$. Specifically, $d$ shifts from a single-prompt update to an implicit dataset-level update $\bar{d} = \mathbb{E}\_{x \sim \rho}[d]$.**
>
> >**[W1-Q1]: How can the authors justify this mismatch between $d$ and $\bar{d}$? What's its influence to the analysis afterwards?**
>
> **W1-Q1-A1**: We appreciate the reviewer's rigorous examination of our derivation. We clarify that the transition from Eq. (11) to Eq. (12) indeed involves an **additional relaxation step via the triangle inequality**, which is explicitly detailed in **Eq. (39) of the Appendix**:
> $$\begin{aligned}
> |\left(\mathcal{L}(\theta\_{\text{old}}+d) - \mathcal{L}(\theta\_{\text{old}})\right)|
> &=|\mathbb{E}\_{x\sim \rho}\left[\mathcal{L}(x;\theta\_{\text{old}}+d) - \mathcal{L}(x;\theta\_{\text{old}})\right]| \newline
> &\le \mathbb{E}\_{x\sim \rho}\left[|\left(\mathcal{L}(x;\theta\_{\text{old}}+d) - \mathcal{L}(x;\theta\_{\text{old}})\right)|\right] \newline
>  &= \mathbb{E}\_{x\sim \rho}\left[\sqrt{2\delta\nabla\_\theta\mathcal{L}(x;\theta)^\top\big|\_{\theta=\theta\_{\text{old}}}F^{-1}(x;\theta\_{\text{old}})\nabla\_\theta\mathcal{L}(x;\theta)\big|\_{\theta=\theta\_{\text{old}}}}\right]\newline
>  &\le \mathbb{E}\_{x\sim \rho}\left[\sqrt{2\delta\mathbb{V}\_{\theta\_{\text{old}}}(r(x,y))}\right] \newline
>  &=\mathbb{E}\_{x\sim \rho}\left[\sqrt{2\delta p\_{\theta\_{\text{old}}}(x)\left(1-p\_{\theta\_{\text{old}}}(x)\right)}\right]
> \end{aligned}$$
>
>
> By applying $|\bar{d}| = |\mathbb{E}\_{x\sim \rho}[d]|\le\mathbb{E}\_{x\sim \rho}[|d|]$, we relax the dataset-level loss term $|\mathcal{L}(\theta\_{\text{old}}+d) - \mathcal{L}(\theta\_{\text{old}})|$, establishing its relationship with instance-level loss term $|\mathcal{L}(x;\theta\_{\text{old}}+d) - \mathcal{L}(x;\theta\_{\text{old}})|$, thereby deriving Eq. (12) from Eq. (11).
>
> **This relaxation implies the resulting upper bound is generally not tight in practice**. However, this bound is **sufficient for our purpose**: it provides **a theoretically grounded estimate of each sample’s optimization potential**, enabling a critical **reweighting mechanism** that down-weights samples with low potential while concentrating computational resources on those with higher potential. In doing so, we substantially enhance **overall training efficiency.**
>
>
>
> >**W1-Q2: Is it accurate to describe the variance term by prompt difficulty? It's more related to the variance. For example, both too difficult and too simple prompt could have zero variance.**
>
> **W1-Q2-A2**: We sincerely appreciate the reviewer’s insightful comment. We fully agree that the term $\sqrt{2\delta p\_{\theta\_{\text{old}}}(x)\left(1-p\_{\theta\_{\text{old}}}(x)\right)}$ **mathematically characterizes the variance of the model’s response accuracy.**
>
> In our work, we framed this using the concept of "prompt difficulty" (i.e., the model's response accuracy $p\_{\theta\_{\text{old}}}$), primarily because its intrinsic relationship with variance: a higher variance indicates that the prompt is of moderate difficulty (neither too hard nor too easy), which consequently implies a greater potential for optimization. We will elaborate on this aspect more carefully in the revised version to eliminate any ambiguity and further enhance the rigor and clarity of our presentation.

---

> ### Author Response · Authors · 2025-11-25
> **Rebuttal to [W2-Q3]**
>
> **[W2]: Unexplained objective and beyond.**
>
> **Section 4.2 proposes optimizing the MSE between actual and theoretical improvements. However, the paper lacks sufficient intuition justifying why minimizing this MSE constitutes a principled curriculum-learning objective under rollout budget constraints.**
>
> >**[W2-Q3]: Why should Eq. (15) be viewed as a valid curriculum-learning objective? What is the intuition?**
>
> **W2-Q3-A3**: We thank the reviewer for this insightful question regarding the intuition behind the objective function in Section 4.2. We clarify the rationale below by distinguishing between the theoretical bounds derived in Section 4.1 and the practical optimization strategy proposed in Section 4.2.
>
> Regarding curriculum learning for reasoning tasks, particularly in algorithms like GRPO, training efficiency is primarily influenced by two critical factors: **problem selection** and the **rollout generation strategy**.
>
> **1. Theoretical Foundation and Problem Selection (Section 4.1)**
> In Section 4.1, we theoretically derive how prompt difficulty (or the variance in response accuracy) constrains the **upper bound of training efficiency**. Based on this derivation, we optimize the prompt sampling distribution to down-weight instances with limited improvement potential and prioritize those likely to accelerate learning. This addresses the first component: **prompt selection**.
>
> **2. Bridging Theory and Practice (Section 4.2)**
> However, the result in Section 4.1 represents an **idealized theoretical upper bound**. In practice, factors such as limited per-prompt rollout budgets and estimation variance of accuracy prevent the model from achieving this bound. Once a batch is sampled, the best we can do is to **approach this theoretical limit as closely as possible**.This is the precise objective of Section 4.2.
> Specifically, in Eq. (15), we have the inequality $|{{\mathcal{L}}}(\hat{\theta}) - {{\mathcal{L}}}(\theta\_{\text{old}})|\le\mathbb{E}\_{x\sim \rho}\left[\sqrt{2\delta p\_{\theta\_{\text{old}}}(x)\left(1-p\_{\theta\_{\text{old}}}(x)\right)}\right]$. By removing the absolute value, we obtain:
> $$
> {{\mathcal{L}}}(\hat{\theta}) - {{\mathcal{L}}}(\theta\_{\text{old}})\ge -\mathbb{E}\_{x\sim \rho}\left[\sqrt{2\delta p\_{\theta\_{\text{old}}}(x)\left(1-p\_{\theta\_{\text{old}}}(x)\right)}\right]
> $$
> (See proof in Appendix Eq. (50)). Consequently, the difference between the actual loss descent and the theoretical bound is always non-negative:
> $$
> {{\mathcal{L}}}(\hat{\theta}) - {{\mathcal{L}}}(\theta\_{\text{old}})-\left(-\mathbb{E}\_{x\sim \rho}\left[\sqrt{2\delta p\_{\theta\_{\text{old}}}(x)\left(1-p\_{\theta\_{\text{old}}}(x)\right)}\right]\right)\ge0
> $$
> Thus, **the objective in Section 4.2 is to minimize the gap between the actual loss reduction and its theoretical upper bound.**
>
> **3. Justification for the MSE Objective**
> Although Eq. (15) is formulated as a Mean-Square Error (MSE) loss, it is mathematically equivalent to:
> $$
> \left({{\mathcal{L}}}(\hat{\theta}) - {{\mathcal{L}}}(\theta\_{\text{old}})-\left(-\mathbb{E}\_{x\sim \rho}\left[\sqrt{2\delta p\_{\theta\_{\text{old}}}(x)\left(1-p\_{\theta\_{\text{old}}}(x)\right)}\right]\right)\right)^2
> $$
> $$
> = \left(\max\left(0, \left({{\mathcal{L}}}(\hat{\theta}) - {{\mathcal{L}}}(\theta\_{\text{old}})-\left(-\mathbb{E}\_{x\sim \rho}\left[\sqrt{2\delta p\_{\theta\_{\text{old}}}(x)\left(1-p\_{\theta\_{\text{old}}}(x)\right)}\right]\right)\right)\right)\right)^2.
> $$
> In other words, **minimizing this MSE loss is equivalent to maximizing the efficiency of the training process.** We selected MSE over L1 loss primarily because **MSE is everywhere differentiable** (which is crucial for theoretical analysis) [1–3] and penalizes large deviations more strongly.
>
> **4. Intuition Behind Incorporating Rollout Budget Constraints**
> Within GRPO-like algorithms, multiple rollouts are required for each prompt. However, not all prompts contribute equally to gradient updates:
> *   **Very easy or very hard prompts** often result in vanishing gradients, providing negligible learning signals.
> *   **Prompts of moderate difficulty** induce substantially higher gradient variance, yielding richer and more informative signals.
>
> To maximize overall training efficiency, the optimal strategy is to **allocate more rollouts to prompts with higher expected training signals**, which will reduce the total variance of the actual training gradients and stabilizes the training process. This is the intuition behind our loss construction, and the direct mechanism by which we address the second key factor: **rollout generation strategy**.
>
>
> **References**
>
> [1] SIGNSGD: Compressed Optimisation for Non-Convex Problems, ICML.
>
> [2] Variance Reduction for Stochastic Gradient Optimization, NeurIPS.
>
> [3] Lower Bounds for Non-Convex Stochastic Optimization, Math. Prog.

---

> ### Author Response · Authors · 2025-11-25
> **Rebuttal to [W2-Q4]-Q7**
>
> >**[W2-Q4]: Since Eq. (15) builds on Section 4.1, it inherits the mismatched issue. How does that affect the soundness of the subsequent results?**
>
> **W2-Q4-A4**: Please refer to our responses to **W1-Q1** and **W2-Q3**.
>
>
>
> >**[W2-Q5]: The analysis simplifies the dependence of advantage estimation on rollout counts, and more critically, the optimal rollout allocation (Eq. 20, 21) seems computationally intractable for LLMs, as computing per-response gradient norms necessitates $N$ backward passes, whereas normal RL algorithms require only a single pass for the batch-averaged gradient.**
>
> **W2-Q5-A5**: We would like to clarify that the per-sample symmetric expression for $\sigma_i$ in Eq. (21) is introduced primarily for **theoretical derivation** to elucidate the variance structure that motivates our rollout allocation strategy; it is **not a form that must be explicitly computed** during actual training. In our implementation, we do **not** perform separate backward passes for individual responses. Specifically, we conduct **a single backward pass per batch**, consistent with standard RL paradigms. From the gradients produced by this single pass, we uniformly compute the gardient norms for all prompts within the batch. This single backward pass to compute the gradient norm does not incur significant overhead.
>
>
> >**[W2-Q6]: What's the time and memory overhead of CurES, especially on estimating? Currently this part is missing in the Experiment Section.**
>
> **W2-Q6-A6**: Regarding memory usage, we are pleased to report that CurES introduces **virtually no additional memory overhead** compared to standard RL algorithms (such as GRPO). Our experiments are conducted on the $8\times H200$ setup, confirming that the memory footprint of CurES is **comparable to that of the GRPO baseline**.
>
> In terms of computational time, we acknowledge a **moderate increase in overhead per step** due to the calculations required for estimating prompt difficulty and gradient norms, as illustrated in Figure 11. Specifically, using the Qwen2.5-Math-1.5B model, the average training time for vanilla GRPO is $20.1 / 545 = 0.037$ h/step, whereas CurES-GRPO incurs $4.5 / 100 = 0.045$ h/step. Thus, each training step introduces an additional overhead of $(0.045 - 0.037) / 0.037 = 22$\% relative to the baseline. However, it is crucial to note that this modest additional cost yields substantial gains in convergence speed. Ultimately, CurES achieves an approximate **$4.5 \times$ speedup in total training time**, thereby significantly enhancing training efficiency.
>
> >**[Q7]: What’s the performance of baseline algorithms with different sample budget? Figure 5 only shows the performance of CurES.**
>
> **[Q7-A7]**: To alleviate the reviewer's concern, we performe experiments using GVM under different sample budget constraints. We have detailed these findings in Figure 12 of the revised paper.
>
> ----
> Finally, we sincerely thank the reviewer for the constructive and insightful comments. We will carefully take all of the feedback into account to further improve the clarity and quality of the manuscript.

---

### Official Review · Reviewer_8JVY · 2025-10-31

**Soundness:** 2
**Presentation:** 2
**Contribution:** 3
**Rating:** 6
**Confidence:** 4

**Summary:**

This work proposes CurES to enhance the training efficiency of Reasoning LLMs. The authors derive a theoretical link between gradient optimization efficiency and both the prompt sampling distribution and rollout quantity allocation. Based on this, CurES adaptively adjusts sampling probabilities and rollout allocations using Bayesian posterior estimation of prompt difficulty. Experiments on mathematical reasoning benchmarks show that CurES achieves faster convergence and higher accuracy than baselinr methods.

**Strengths:**

- The work addresses a practically relevant problem—improving the training efficiency of reasoning LLMs.
- CurES reallocates both prompt sampling probabilities and rollout quantities to enhance the training efficiency.
- Many of the derived equations have closed-form solutions, which makes computation and optimization more tractable.
- The presentation of several figures is excellent. For example, the distributions of rollout quantities and estimated accuracies in Figures 3 and 4 are clearly illustrated, providing intuitive insights into the CurES’s behavior throughout the training process.

**Weaknesses:**

- In Eq. (12), the $\leq$ relation does not seem to strongly support the argument. Since the method aims to maximize the right-hand side (RHS), having a higher upper bound on the RHS does not necessarily imply a larger loss change in LHS. A $\geq$ lower bound would better motivate and validate the theory and the method.
- The training paradigm is restricted to iterative training across pre-divided dataset subsets, which limits generality. The way the dataset is partitioned may influence the algorithm’s performance. Moreover, each inner iteration requires rolling out every prompt to estimate the initial difficulty distribution, which introduces additional computational overhead.
- The methodology presentation lacks clarity. While the paper provides detailed explanations of the gradient theory and the estimation of prompt difficulty, it does not clearly describe how sampling and budget allocation are actually conducted. Before seeing Algorithm 1 in the Appendix, it remains unclear how the algorithm operates in practice.
- The paper could include more ablation studies to isolate the individual contributions of prompt sampling versus rollout allocation.
- The notation introduced in Section 5.3 (Line 460) needs revision for clarity and consistency. "pre-sampling scale (N)": This term is ambiguous. Moreover, it conflicts with the earlier definition of N as the "total rollout budget" on Line 224. "training-phase sample budgets coefficient (n)": This term is imprecise, and the symbol n is not defined previously in the text.

**Questions:**

- Line 350: "an assignment of rollout quantities under a total sample budget of 8×1024". Does it mean that the batch size for training CurES is set to 1024?

---

> ### Author Response · Authors · 2025-11-25
> **Rebuttal to W1-W2**
>
> **Dear Reviewer 8JVY**:
>
> We extend our sincere gratitude to the reviewer for the constructive comments and suggestions. Below is our response to your concerns:
>
> **[W1] Eq. (12)’s $\le$ relation is unconvincing, as maximizing an upper bound does not guarantee LHS improvement. A lower bound would better validate the theory.**
>
> **W1-A1**: We thank the reviewer for this insightful suggestion. We fully acknowledge that employing a conservative lower bound estimation would ideally offer a stronger guarantee for the model's actual gradient updates. However, **establishing a strict lower bound for $|L(\theta_{\text{old}} + d) - L(\theta_{\text{old}})|$ is mathematically non-trivial, as it fundamentally requires modeling the complex correlations among different prompts** (since their gradients often interact). Due to this inherent uncertainty and complexity, guaranteeing such a lower bound in practice proves challenging. Moreover, incorporating prompt correlations may render it difficult to apply effectively in experimental settings.
>
> In our approach, the derived upper bound for $|L(\theta_{\text{old}} + d) - L(\theta_{\text{old}})|$ represents the **theoretical maximum optimization cap** under the given constraints.  While this does not strictly guarantee a large loss change in $|\Delta L|$, it naturally provides a critical **screening mechanism**: samples with small upper bounds are theoretically incapable of inducing substantial loss changes. Conversely, **only samples with sufficiently large upper bounds possess the potential for "rapid updates."** In this sense, the upper bound serves as a **"necessary condition for achieving significant improvements."** This rationale directly motivates our redistribution strategy in Eq. (14), which **prioritizes exploration among samples with greater theoretical potential** under an entropy maximization constraint.
>
> To further bridge the gap between theory and practice, **we analyse the alignment of actual training with this theoretical bound in Section 4.2**. Specifically:
> *   **Section 4.1** derives the **theoretical upper bound as a prerequisite** for prompt reweighting.
> *   **Section 4.2** (via Eq. (15) and Eq. (16)-Eq. (21)) **analyzes and minimizes the discrepancy between actual and theoretical gradient updates** by dynamically adjusting the number of rollouts.
>
> Thus, these two sections are designed to be **complementary**: one establishes the potential, while the other aims to bring actual updates as close as possible to that potential.
>
>
>
> **[W2]: The reliance on iterative subset training limits generality and introduces partition sensitivity, while full rollouts for estimation incur significant computational overhead.**
>
> **W2-A2**: We respectfully clarify that **our approach is agnostic to the specific dataset splitting method employed.** The partitioning scheme adopted in our experiments simply aligns with the setup in GVM [1] to **ensure reproducibility and fairness in our comparative analysis.**
>
> The rationale behind dividing the dataset and utilizing iterative training is to **enhance the estimation precision of the current model’s answer accuracy (i.e., prompt difficulty).** This strategy offers two primary advantages:
> 1.  It allows for a more concentrated sampling distribution within each iteration, thereby **increasing the average sampling frequency per prompt and mitigating estimation bias** arising from delayed sampling.
> 2.  It facilitates **timely updates to difficulty statistics** by discarding outdated information. This ensures that estimates remain **aligned with the current model’s capabilities**, effectively reducing the impact of **distribution shifts induced by policy updates.**
>
> While we acknowledge that this strategy introduces **moderate additional computational overhead, it provides more reliable and policy-aligned difficulty estimates, ultimately enhancing the overall training efficiency and stability of the training process.**
>
>
> **References**
>
> [1] Optimizing Chain-of-Thought Reasoners via Gradient Variance Minimization in Rejection Sampling and RL

---

> ### Author Response · Authors · 2025-11-25
> **Rebuttal to W3-W5 & Q1**
>
> **[W3]: The methodology lacks clarity on practical sampling and budget allocation. Despite detailed theory, the algorithm's operation remains ambiguous without referring to Algorithm 1 in the Appendix.**
>
> **W3-A3**: We appreciate the reviewer’s constructive feedback regarding the clarity of our presentation. We clarify that our proposed algorithm operates as a **two-stage procedure**:
>
> 1.  **Parameter Estimation Stage**: We perform a limited number of rollouts (parameter estimation sample
> size $N'$, set to 4 in our experiments) to **gauge the prompt difficulty and gradient variance relative to the current model policy.**
> 2.  **Training Stage**: Leveraging these initial estimates, we proceed to the main training phase. In this stage, we first sample $m$ prompts according to Eq. (14) to form a prompt batch, and then collect a total of $N$ rollouts for this batch following our rollout allocation strategy specified in Eq. (20). This implies that, on average, each prompt receives $n$ rollouts, satisfying $N = m \times n$.
>
>
> To address your concern, **we have incorporated a more detailed description of the algorithm’s implementation in the revised manuscript** to significantly enhance the overall clarity and readability. Please refer to the updated version.
>
>
>
> **[W4]: Additional ablation studies are needed to isolate the specific contributions of prompt sampling versus rollout allocation.**
>
> **W4-A4**: To address this, we conducted ablation studies on the Qwen2.5-Math-1.5B model to **isolate the individual contributions of the two core components of CurES: prompt difficulty–based sampling** and **rollout allocation**. The results are presented below:
>
> | Method                             | MATH500 | GSM8K | GAO23 | MINERVA | OLYM  | AIME24 | AIME25 | AMC23 | Avg.  |
> | ---------------------------------- | ------- | ----- | ----- | ------- | ----- | ------ | ------ | ----- | ----- |
> | Qwen2.5-Math-1.5B                  | 40.20   | 43.90 | 25.19 | 11.40   | 21.04 | 1.67   | 1.67   | 14.84 | 20.00 |
> | +CurES-GRPO                        | 77.20   | 85.97 | 51.43 | 31.62   | 37.33 | 13.33  | 10.42  | 52.19 | 44.94 |
> | +CurES-GRPO w/o Rollout Allocation | 76.20   | 84.38 | 49.61 | 29.04   | 36.30 | 13.33  | 9.38   | 51.41 | 43.71 |
> | +CurES-GRPO w/o Prompt Sampling    | 77.00   | 84.00 | 50.25 | 29.04   | 36.59 | 13.33  | 11.46  | 50.94 | 44.08 |
>
> These empirical findings indicate that **both components contribute significantly to the final performance improvement**. Specifically, **removing either difficulty-based sampling or rollout allocation leads to a measurable degradation in overall accuracy**, thereby confirming their **complementary roles** within the CurES framework.
>
> **[W5]: The notation in Section 5.3 requires revision due to ambiguity and inconsistency. Specifically, "pre-sampling scale (N)" conflicts with the earlier definition of N as total budget, while "training-phase sample budgets coefficient (n)" is imprecise and introduces an undefined symbol.**
>
> **W5-A5** We sincerely thank the reviewer for this careful observation and apologize for the confusion caused by the notation error regarding "pre-sampling scale ($N$)", which was intended to be $N'$. To clarify:
>
> *   **$N'$ actually refers to the number of rollouts performed during the parameter estimation stage**, and we have revised the manuscript to clarify this parameter and now refer to it as the parameter estimation sample size；
> *   The **rollout budget coefficient $n$** corresponds to the rollout quantity used in the baseline GRPO setting, **representing the average number of rollouts allocated per prompt.**
> *   Consequently, the **total rollout budget is defined as $N = m \times n$**, where $m$ denotes the prompt batch size.
>
> We have **thoroughly revised the definitions and notation expressions in the updated manuscript** to ensure consistency and clarity.
>
> ----
>
> **Q1: Line 350: "an assignment of rollout quantities under a total sample budget of 8×1024". Does it mean that the batch size for training CurES is set to 1024?**
>
> **A1**: We confirm that this parameter refers to our training batch size, which is explicitly set to 1024 in our experimental setup.
>
> ----
>
> Finally, we sincerely thank the reviewer for the constructive and insightful comments. We will carefully take all of the feedback into account to further improve the clarity and quality of the manuscript.

---

### Official Review · Reviewer_uxBd · 2025-11-01

**Soundness:** 3
**Presentation:** 3
**Contribution:** 3
**Rating:** 6
**Confidence:** 2

**Summary:**

The paper introduces CurES that selects prompts and determines the number of rollouts associated with each prompt during post-training based on the difficulty of the prompt in order to improve training efficiency.

**Strengths:**

The method is based on a theoretical analysis of the relationship between training dynamics and difficulty of prompts.

The experiments demonstrate that CurES improves average accuracy in mathematical reasoning across a range of model sizes compared with alternative curriculum approaches, namely Speed-RL and GVM (the quantitative results in Table 1).

**Weaknesses:**

What the experiments related to the sampling behavior (in Section 5.2) are supposed to demonstrate is not clear. In Figure 4, “the distribution of estimated accuracy … becomes more concentrated around higher values,” but wouldn’t we expect this to be the case for any learning method?

Although Speed-RL and GVM select the number of rollouts in a more straightforward manner, reporting their rollout numbers in Figure 3 would help highlight the difference of CurES.

In general, the sampling behavior analysis and the efficiency analysis in Sections 5.2 and 5.3 are only performed for CurES. Contrasting these results against Speed-RL and GVM can make these analyses more informative.

How do we arrive at the conclusion about Figure 5? Figure 5 doesn’t report the increase in computational cost with respect to N and N. How do we conclude they do not yield a proportional increase in performance? Assuming that’s true, how does this relate to the CurES’s efficiency?

**Questions:**

In Figure 6, does one step for CurES-GRPO/RPP have the same computational cost as vanilla GRPO/RPP? Would the results have been different if the x-axis was computational cost (for instance measured in FLOPS) rather than steps?

---

> ### Author Response · Authors · 2025-11-25
> **Rebuttal to W1-W4**
>
> **Dear Reviewer uxBd**,
>
> We extend our sincere gratitude to the reviewer for the constructive comments and suggestions. Below is our response to your concerns:
>
>
>
> >**[W1] The sampling experiments (Section 5.2) lack clarity. Why is the concentration of estimated accuracy at higher values noteworthy, as it seems generally expected?**
>
> **W1-A1:** We appreciate the reviewer’s insightful comment. While we agree that the convergence toward higher accuracy is a general trend for most learning methods. Section 5.2 aims to illustrate **the evolution of the accuracy distribution** and, crucially, to demonstrate that **medium-difficulty samples become scarce** in the true distribution as training progresses. This implies that maintaining uniform sampling would cause **the effective samples available for gradient updates to diminish**, thereby **lowering training efficiency**. These findings underscore **the necessity of redistributing the prompt sampling distribution (Eq. (14))**.
>
>
> >**[W2] Reporting Speed-RL and GVM rollout numbers in Figure 3 would highlight CurES's difference.**
>
> **W2-A2:** We would like to clarify that Speed-RL primarily filters prompts without modifying the rollout quantity, implying that **Speed-RL still employs a uniform sampling strategy**. Regarding **GVM’s allocation of varying rollout numbers**, these results are now illustrated in **Figure 13 of our revised paper**.
>
> It is worth noting that **the theoretical analysis of GVM is conducted from the perspective of EM estimation**. Given that, under the rejection sampling framework, **the variance for the $i$-th prompt is approximately proportional to $\frac{1}{n_i p_i}$** (where $n_i$ denotes the number of rollouts allocated to the $i$-th prompt, and $p_i$ represents the accuracy rate of the response to the $i$-th prompt), **GVM allocates more rollouts to prompts with lower accuracy rates**. This strategy allows it to **minimize the overall gradient variance while maintaining a fixed total budget**.
>
>
> >**[W3] Comparing CurES with Speed-RL and GVM in Sections 5.2 and 5.3 with the sampling behavior analysis and the efficiency analysis would make the analysis more informative.**
>
> **W3-A3**: Regarding the sampling behavior analysis, we kindly refer the Reviewer to **W2**. Additionally, we present the results of the efficiency analysis in **Figure 11 of our revised paper**. These results demonstrate that **CurES consistently accelerates training efficiency compared with Speed-RL and GVM**.
>
>
>
>
> >**[W4]: Conclusion about Figure 5 and how does this relate to the CurES’s efficiency?**
>
> **W4–A4:** Figure 5 is presented to demonstrate the **robustness of CurES to the hyperparameters $(N', n)$**, showing that the model consistently achieves performance gains across various configurations.
>
> Our algorithm operates in two phases: the **parameter estimation stage** and the **large-scale training stage**. Specifically, $N'$ represents the number of rollouts used to estimate the initial prompt difficulty and gradient variance, whereas $n$ denotes the average rollouts per prompt during training (defined as $n = N / m$, where $m$ is the prompt batch size, and $N$ is the total rollout budget).
>
> Consequently, **variations in $N'$ isolate the estimation phase, while changes to $n$ specifically affect the training phase.** The computational costs associated with these different $(N', n)$ settings are detailed below:
>
> **Parameter Estimation Stage**
>
> | $(N',n)$ Setting | Avg. Time Cost |
> | :--------------- | :--------------------------- |
> | **(4, 8)**       | 0.258h                       |
> | **(8, 8)**       | 0.575h                       |
> | **(16, 8)**      | 1.033h                       |
>
> **Large-scale Training Stage**
>
> | $(N',n)$ Setting | Avg. Time Cost per Step |
> | :--------------- | :---------------------- |
> | **(4, 8)**       | 0.137h                  |
> | **(4,16)**       | 0.275h                  |
> | **(4,32)**       | 0.548h                  |
>
> While increasing $N'$ and $n$ inevitably raises training overhead, the final performance remains consistent across various settings. Our results indicate that even a modest configuration, such as $(4, 8)$, is sufficient to satisfy training objectives. Consequently, **CurES proves to be computationally efficient, as it does not require extensive resources for parameter estimation, validating the use of small $N'$ and $n$ values in practical applications.**

---

> ### Author Response · Authors · 2025-11-25
> **Rebuttal to Q1**
>
> >**[Q1]: Do CurES and vanilla steps have equal cost, and would results differ if plotted against computational cost in Figure 6?**
>
> **Q1-A1**: **As illustrated in Figure 11 of our revised paper**, we use the x-axis to represent GPU hours and redrawn **the performance and training time curves**. Specifically, for the Qwen2.5-Math-1.5B model, the average training overhead per step for vanilla GRPO is **$0.037$ h/step**, whereas our CurES-GRPO algorithm incurs **$0.045$ h/step**. Although estimating problem difficulty and gradient norms introduces **an additional overhead of approximately $22$\% per step**, this modest cost yields **substantially greater training acceleration**. Overall, our CurES algorithm achieves **approximately $4.5 \times$ speedup in total** compared to vanilla GRPO, thereby **significantly enhancing training efficiency**.
>
> ---
> Finally, we sincerely thank the reviewer for the constructive and insightful comments. We will carefully take all of the feedback into account to further improve the clarity and quality of the manuscript.

---

### Comment · Area_Chair_tac3 · 2025-11-26
**Reminder of the Update PDF and Discussions**

Hi Author,

Remember to update the pdf to its latest version by incorporating all latest revisions or new results. This will work as the crucial evaluation.

Best,
AC

Hi Reviewers,

Remember to attend the author-reviewer discussions before the deadline Dec. 3rd. Your feedback is also important.

Best,
AC

---

> ### Author Response · Authors · 2025-11-27
>
> **Dear Area Chairs and Reviewers,**
>
> We have updated the PDF to the latest version, incorporating all recent revisions and new results to improve the clarity of our paper. We sincerely appreciate the time and effort dedicated to reviewing our work, and we remain available to answer any further questions.
>
> Best,
>
> **Authors of Submission 16902**

---

### Author Response · Authors · 2025-11-30
**Summary of Rebuttal**

**Dear Area Chair and Reviewers,**

We sincerely thank you for the thorough engagement with our work. The review process has been highly constructive, leading to significant improvements in the clarity, scope, and validation of our paper. We have incorporated all revisions into the latest PDF.

Below, we synthesize the consensus on the paper's strengths, summarize the major common weaknesses identified, and detail the key steps taken in the rebuttal to address these concerns.

---

> **I. Paper Strengths and Core Contributions**

1.  **Principled Theoretical Foundation:** The core strength of our work lies in providing a **systematic and theoretical investigation** into training efficiency. We derive a principled link between gradient optimization efficiency and the two key factors: **prompt sampling distribution** and **rollout quantity allocation**. This analysis aligns with intuition (e.g., prompts of intermediate success probability are the most informative) and is successfully converted into the practical CurES algorithm.
2.  **Superior Performance and Efficiency:** CurES is an efficient training method that achieves **faster convergence and higher final accuracy** compared to existing methods. Experiments demonstrate that CurES **exceeds the best prior sample-efficient methods by +2.12 points on average** across eight mathematical reasoning benchmarks.
3.  **Practical Relevance and Intuitiveness:** Our work addresses the **practically relevant problem** of improving the training efficiency of reasoning LLMs. Furthermore, the presentation of results (e.g., distributions of rollout quantities in Figures 3 and 4) was praised for providing **intuitive insights** into the algorithm's behavior.

> **II. Common Weaknesses and Major Concerns**

1.  **Algorithmic Clarity [8JVY, 9BeW]:** The actual implementation details, particularly the **two-stage process** involving parameter estimation (pre-rollout) and budget allocation, were initially unclear without referring to the Appendix.
2.  **Ablation Studies [8JVY, 9BeW]:** Reviewers consistently requested more detailed ablation studies to **quantify the independent contributions** of the two core components: prompt sampling and rollout allocation.
3.  **Computational Overhead [uxBd, Lzg2]:** The manuscript lacked clear details on the **time and memory overhead** introduced by estimating prompt difficulty and gradient norms ($\sigma_i$), and whether efficiency results would hold when plotted against GPU computational cost.
4.  **Limited Experiments [9BeW]:** Experiments were initially limited to the **Qwen2.5-Math models**, raising questions about the method's **robustness and generalizability** to diverse model architectures (e.g., LLaMA, Phi).

> **III. Key Revisions and Responses**

| Concern | Author Response and Key Revision |
| :--- | :--- |
| **A. Algorithmic Clarity** | We clarified the **two-stage procedure**: 1) A **Parameter Estimation Stage** (using limited rollouts $N'=4$) to gauge prompt difficulty, followed by 2) the **Training Stage**, where prompts are sampled using Eq. (14) and rollouts allocated using Eq. (20). This detailed explanation and an integrated Algorithm 1 are now in the main text. |
| **B. Ablation Studies** | **New ablation studies** were conducted on Qwen2.5-Math-1.5B, isolating the contributions of sampling and allocation. Results showed that **removing either component led to a measurable degradation in accuracy** (e.g., $44.94 \to 43.71$ or $44.08$), confirming their **complementary and integral roles** in the CurES framework. | |
| **C. Efficiency** | **Computational cost was quantified:** CurES introduces **virtually no additional memory overhead**. While estimating parameters adds a **moderate overhead of approximately 22% per step** ($0.045 \text{ h/step}$ vs. vanilla GRPO's $0.037 \text{ h/step}$), this modest cost yields a significant **4.5x overall speedup** in total training time. Results have been replotted against GPU hours to confirm efficiency. |
| **D. Generality** | To address limited model diversity, **new experiments** were performed on the **LLaMA and Phi model families**. The results demonstrate that **CurES-GRPO consistently outperforms vanilla GRPO** across these diverse architectures, confirming the method's robustness and generalizability. |
| **E. Baselines** | We added experimental comparisons against relevant online methods **DAPO and MoPPS**, showing CurES achieves superior overall performance. We also clarified that our method is **gradient-driven**, using difficulty estimation only as a supporting component to instantiate the theoretical sampling and allocation formulas, thus distinct from MoPPS. |

---
We believe these revisions and supplementary results fully address the reviewers' concerns, solidifying the contribution and technical rigor of the CurES methodology. We are available for any further clarification if needed.

Sincerely,

**Authors of Submission 16902**

---

### Meta-Review · Area_Chair_9SRm · 2026-01-09

**Summary:**

The paper examines a curriculum learning strategy for LLM reasoning, specifically by deciding on an optimal allocation strategy.

**Reviewer Concerns:**

The main weakness is that the theory relies on loose relaxations and upper-bound arguments, so the “principled” derivation is not fully rigorous. In addition, efficiency and generality are not conclusively demonstrated, since compute costs, baseline comparisons, and non-math settings are limited.

**Reviewer Scores:**

All reviews recommend acceptance.

---

### Decision · Program_Chairs · 2026-01-26

Accept (Poster)